# Understanding Model Selection for Learning in Strategic Environments

**Tinashe Handina** *
Computing + Mathematical Sciences
California Institute of Technology
Pasadena, CA 91125
thandina@caltech.edu

**Eric Mazumdar**
Computing + Mathematical Sciences
California Institute of Technology
Pasadena, CA 91125
mazumdar@caltech.edu

## Abstract

The deployment of ever-larger machine learning models reflects a growing consensus that the more expressive the model class one optimizes over—and the more data one has access to—the more one can improve performance. As models get deployed in a variety of real-world scenarios, they inevitably face strategic environments. In this work, we consider the natural question of how the interplay of models and strategic interactions affects the relationship between performance at equilibrium and the expressivity of model classes. We find that strategic interactions can break the conventional view—meaning that performance does not necessarily monotonically improve as model classes get larger or more expressive (even with infinite data). We show the implications of this result in several contexts including strategic regression, strategic classification, and multi-agent reinforcement learning. In particular, we show that each of these settings admits a Braess' paradox-like phenomenon in which optimizing over less expressive model classes allows one to achieve strictly better equilibrium outcomes. Motivated by these examples, we then propose a new paradigm for model selection in games wherein an agent seeks to choose amongst different model classes to use as their action set in a game.

## 1 Introduction

Machine learning—and deep learning in particular— has already demonstrated enormous potential to enable new services across a wide spectrum of everyday life. Examples range from chatbots [1], to hiring [2], and content moderation [3]. Driving this proliferation is the fact that the increasing availability of compute resources coupled with the abundance of data provided by internet-scale datasets allows one to train larger and larger models while monotonically improving performance [4, 5, 6]. Despite signs of diminishing returns, this consensus has broadly held true. Machine learning algorithms tend to follow a monotonic scaling law: with more compute and data, one can train more expressive models and eke out performance gains.

As algorithms are deployed into real-world scenarios, however, they will inevitably come into contact with some form of strategic decision-making—whether that be in the form of adversarial agents attempting to manipulate the output of the algorithm [7], gig-workers taking actions to enforce better working conditions from learning-powered platforms [8], or more broadly individuals whose goals are misaligned with those of the algorithm [9].

Reflecting this reality, recent years have seen a surge in research seeking to understand the effects of strategic decision-making on learning algorithms. Two sub-fields in particular include *strategic* classification [10]—in which one seeks to learn a classifier or predictor in the presence of agents

---

*Correspondence to Tinashe Handina <thandina@caltech.edu>.

38th Conference on Neural Information Processing Systems (NeurIPS 2024).

who strategically manipulate data— and multi-agent reinforcement learning [11]— in which agents attempt to learn optimal decision-making policies in the presence of other learning agents. Both of these domains draw on ideas from game theory and economics to understand how to design algorithms for strategic settings.

In these different regimes, a common refrain is that the presence of strategic interactions invalidates many of the foundational assumptions underlying many machine learning algorithms. For example, strategic interactions result in highly non-stationary environments for multi-agent reinforcement learning (MARL) [11] and seemingly innocuous design decisions like stepsize choices and gradient estimators have been shown to give rise to qualitatively different outcomes in strategic classification [12]. Despite these works, our understanding of model class selection in strategic settings remains underdeveloped. To that end, in this paper, we consider the following question:

*How do strategic interactions affect the relationship between model class expressivity and equilibrium performance?*

**Contributions:** We show through simple theoretical models, illustrative examples, and experiments that strategic interactions can yield a non-trivial relationship between model class expressivity and equilibrium performance. In particular, we show how—even in highly structured regimes in which one has full access to the underlying data distribution—strategic interactions can result in a Braess' paradox-like phenomenon: the larger and more expressive the model class a learner optimizes over, the lower their performance at equilibrium.

To understand why this is possible, we make links with the literature in economics on comparative statics, which seeks to understand how the equilibria of games vary with exogenous factors. We show that *even* in convex games with a unique equilibrium, if the equilibrium is *not* Pareto optimal (i.e., there exists coordinated deviations that improve the utilities of both players), then there always exists a *unilateral* restriction of one's action set over which one could have played and had a better equilibrium outcome. Conversely, we show that if an equilibrium is Pareto optimal (which encompasses not only traditional optimization but also adversarial games), performance at equilibrium will tend to scale monotonically with respect to model class expressivity. To make this result concrete, we give examples of strategic regression, strategic classification, and MARL in which *reverse* scaling occurs.

Our result suggests that—if the model will be deployed into a strategic environment— the choice of model class should be treated as a strategic action. Following up on this observation, we formulate a problem of model-selection in games. Whereas learning in games traditionally takes the action set for a player as given, we propose a new formulation in which a player has a number of action sets to choose from and must find the one that yields the best payoff. As a proof-of-concept, we provide an algorithm to identify the best set in a class of structured games.

## 1.1 Related work

Before describing our model and results, we comment on related work in both machine learning and economics.

**Scaling laws in Machine Learning:** Within statistical machine learning, the study of scaling laws is motivated by the task of choosing a sufficiently expressive class of models to optimize over a given dataset [13]. Non-monotonicity of scaling laws emerged classically due to overfitting [14]—in which the model is too expressive relative to the size of the data set, which can degrade performance at deployment. Such problems are fundamentally linked to understanding the behavior of the empirical risk as one optimized over larger and larger model classes [15].

Deep learning upended this way of thinking with the development of a theory for generalization beyond what is called the threshold of interpolation[16]. More recently, work has investigated how the performance of large language models scales with expressivity (measured in the number of parameters) [4, 5, 6] and the size of the dataset it is trained on [17]. In each of these works, the scaling laws increase monotonically in both the amount of data and expressivity.

In our paper, we sidestep issues of sample complexity (i.e., dataset size) to isolate the interplay between expressivity and strategic interactions. While the minimum of the population risk for supervised learning monotonically decreases because optimizing over a larger space can only improve performance, we show that the population risk is non-monotonic in strategic environments. Thus, the

phenomenon that we highlight holds even without consideration of sample sizes or generalization errors and adds to a growing body of literature on the difficulties of learning in game theoretic environments.

**Learning in strategic environments:** Recent years have seen a surge of interest in understanding the effects of strategic interactions on learning algorithms. Some of the most relevant areas of interest are strategic classification [10] and performative prediction [18], adversarial machine learning [7], and multi-agent reinforcement learning [11]. A unifying theme across these areas is the integration of ideas from game theory into problems of machine learning, wherein one seeks to learn an optimal model given the presence of strategic agents who may themselves be learning. For example, papers on strategic classification [19], strategic regression [20], and participation dynamics [21, 22] all analyze games in which a learner deploys learning algorithms in game-theoretic environments. Similarly, work on MARL naturally builds upon the foundation of Markov games [23, 24].

One can view all of these problems as an instance of learning in games [25]—which has seen a resurgence in the machine learning literature in recent years due to these connections [26, 27]. In this paper, we adopt, in particular, the framework of continuous games [28] in which players' action sets can be compact convex subsets of $\mathbb{R}^n$.

In this class of games, recent work has made clear that learning can be much more complex than in stationary environments— with non-trivial consequences including instability and convergence to cycles and chaos when using gradient-based learning [29], small design choices like stepsizes and gradient estimators leading to different equilibria [12], and strategic manipulations allowing for better causal discovery [30].

The question of whether our current understanding of scaling laws holds in these environments is still relatively understudied. Recent empirical work has shown that scaling laws in zero-sum MARL mirror those for deep reinforcement learning and deep learning more generally [31]. Most relevant to our work is a recent paper that studied the non-monotonicity of users' social welfare as firms deploy larger and larger models [32]. The paper considers an environment in which multiple firms compete over a set of users and analyzes the welfare of the users as all firms choose more complex models. They show through a simple model and extensive experiments that if all firms use larger models, the users' welfare can decrease. In this paper, we formulate a more general model that encompasses their interaction and more general problems of strategic classification and MARL. We take an orthogonal track, which is to ask whether it is rational for self-interested learners to *unilaterally* restrict the expressivity of their models in strategic settings. We show that this is indeed the case under certain conditions.

**Changing equilibrium outcomes in game theory:** Finally, we would be remiss if we did not discuss the large body of work in economics that studies changes in equilibrium outcomes in games. A similar phenomenon to the one we highlight is the well-known Braess' paradox in strategic routing [33] in which one can add a road to a network and increase congestion. Even more related is the *informational* Braess' paradox [34] in which more information over the network can yield worse equilibrium outcomes for agents in routing games. Many classic works in dynamic game theory have also highlighted the unintuitive ways information and statistical estimation affect equilibrium outcomes in games [35, 36].

More generally, a large body of work in economics studies comparative statics—i.e., how equilibrium payoffs change as exogenous variables are changed [37]. The literature has mostly been concerned with deriving conditions under which payoffs change monotonically in the exogenous variables, a field known as a *monotone* comparative statics [38]. Our work can be seen as an attempt to understand these ideas in the context of strategic machine learning.

## 2   Preliminaries

To understand the dependencies between strategic decision-making and model complexity, we examine different strategic environments. In our model, the learner has access to an ordered set of model classes $\mathbb{A}$, which are all subsets of one large class $\Omega$.

**Definition 2.1.** A set of model classes $\mathbb{A} = \{\Theta_k\}_{k=1}^N$ is *ordered* if for all $\Theta_i, \Theta_j \in \mathbb{A}$, if $i < j$ it implies that $\Theta_i \subseteq \Theta_j$

The set $\mathbb{A}$ may be a set of nested policy classes in MARL or a set of neural network architectures of increasing size for strategic classification. Importantly, the model classes have monotonically increasing expressivity when measured in classic notions of expressivity like V-C dimension [15].

Before engaging with the strategic environment, the learner chooses a model class $\Theta_i \in \mathbb{A}$ over which to optimize. In some instances, we refer to model classes as action spaces, and we use these two terms interchangeably. The selection of a model class fixes the optimization problem the learner will then attempt to solve through interactions with the environment. We model interactions with the environment as a two-player game and assume that players find equilibrium outcomes. We assume the learner has a loss function $f_l : \Omega \times \mathcal{E} \to \mathbb{R}$ which they seek to minimize that also depends on the action of the environment. Similarly, the environment will have a loss function $f_e : \Omega \times \mathcal{E} \to \mathbb{R}$. Here $\Omega$ is the learner's action space whilst $\mathcal{E}$ is the environment's action space.

To model different strategic interactions, we make different assumptions on the nature of the game played and the equilibrium outcomes. We focus on four types of strategic environments: Stationary Environments where the environment actor only has a single action, Stackelberg Environments where the Learner leads, Stackelberg Environments where the learner follows, and General Nash Environments. We provide a concrete description of each of these environments in Appendix A.1.

In the next section, we analyze General Nash environments and show that payoffs do not necessarily monotonically increase in expressivity. In Appendix A.2, we provide a set of theoretical results that show how equilibrium payoffs *are* monotonically increasing in expressivity in Stationary and Stackelberg environments where the learner leads. We then show in Appendix A.3 how payoffs also do not necessarily monotonically increase in expressivity in Stackelberg environments where the learner follows.

# 3   Non-Monotonic Scaling of Performance in Nash Settings

In this section, we present our investigation of the relationship between model class expressivity and equilibrium performance in Nash setting. We show how *even* under strong assumptions on the regularity of the game, there always exists a way for a player to restrict their model class, resulting in a game with a Nash equilibrium that has a lower loss if the original Nash equilibrium is not Pareto optimal (i.e., the game is not zero-sum or strategically zero-sum). We note that this is a negative result which is an existence proof. To concretely establish this phenomenon, we illustrate through examples in multi-agent reinforcement learning and strategic classification how in settings where these assumptions are relaxed, this phenomenon still exhibits itself.

To prove our main result, we assume the two-player game is strongly monotone on the space $\Omega \times \mathcal{E} \subset \mathbb{R}^n$.

**Definition 3.1.** A two-player game is $\mu$-strongly monotone if the generalized gradient operator $F : \Omega \times \mathcal{E} \to \mathbb{R}^n$ given by:

$$F(x) = \begin{bmatrix} \nabla_\theta f_l(\theta, e) \\ \nabla_e f_e(\theta, e) \end{bmatrix} \quad \text{where: } x = (\theta, e),$$

satisfies:

$$\langle F(x) - F(x'), x - x' \rangle \geq \mu \|x - x'\|^2 \ \forall \, x, x' \in \Omega \times \mathcal{E}$$

A strongly monotone game is a convex game [28]. Implicitly, it assumes that the two players' losses are $\mu$ strongly convex in their own action and makes a further assumption on the interaction between players' actions [27]. The assumption of strong monotonicity ensures that there is always a unique Nash equilibrium and that issues of multiple equilibria do not arise.

This assumption is once again made to isolate the phenomenon of interest. In the case with multiple Nash equilibria we believe that it is possible to have different equilibrium outcomes exhibiting different scaling behavior—though we leave such analyses for future work. We remark that we make these assumptions for illustrative purposes and that many of our numerical experiments show the same result under milder game structures.

On top of the assumption of strong monotonicity we require several smoothness conditions on the players' objectives as well as an assumption that their interaction is not trivially zero at Nash.

**Assumption 3.2.** Assume the game defined on $f_e$ and $f_l$ is strongly monotone on $\Omega \times \mathcal{E}$. Further assume that

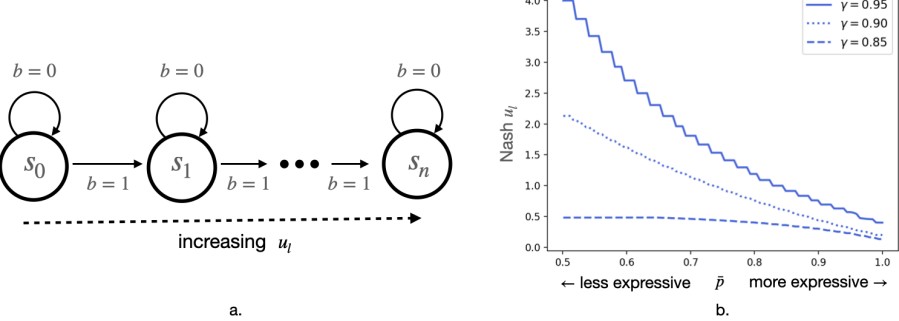

a.                                    b.

Figure 1:  (a.) A visual description of a 2-player Markov game in which the learner can unilaterally increase their payoff by restricting the expressivity of their policy class. (b.) the payoff of the learner at Nash in a 50-state version of this Markov game as their policy class is restricted to take the form $\pi_l(s) = [p, 1-p]$ in all states $s$ for $p \in [1 - \bar{p}, \bar{p}]$ for different discount factors (assumed to be the same for both players). In all cases, we see the learners' payoff broadly *increase* at Nash as they optimize over smaller policy classes.

1. $f_l$ and $f_e$ are jointly convex in $\theta$ and $e$.

2. The gradient mappings, $\nabla f_l$ and $\nabla f_e$ exist and are well defined for all $(\theta, e)$. Furthermore, the gradient mappings are $L$-Lipschitz continuous in the joint action space.

3. The Nash equilibrium $\theta^* \in \Theta$ is on the interior of $\Theta$ with $\nabla_\theta BR_e(\theta^*) \neq 0$.

To show how the restriction of a model class yields a decrease in loss in a large class of games, we leverage the idea that in many games, a Nash equilibrium is not necessarily a Pareto optimal point [39].

**Definition 3.3.** A point $(\theta, e) \in \Omega \times \mathcal{E}$ is Pareto-optimal, if there *does not* exist $(\hat{\theta}, \hat{e})$ such that $f_l(\hat{\theta}, \hat{e}) < f_l(\theta, e)$ and $f_e(\hat{\theta}, \hat{e}) \leq f_e(\theta, e)$ [2]

Given these assumptions, we prove the following theorem. For ease of exposition, we defer the proof to Appendix B.1.

**Theorem 3.4.** *For a two-player monotone game $G$ on $\Theta \times \mathcal{E}$ which satisfies Assumption 3.2, if the unique Nash equilibrium in $\Theta \times \mathcal{E}$, $(\theta^*, e^*)$, is not Pareto optimal then there exists a restriction of the learner's model class (i.e., a set $\Theta' \subset \Theta$) such that the restricted game $G'$ on $\Theta' \times \mathcal{E}$ admits a Nash equilibrium $(\theta', e')$ with: $f_l(\theta', e') < f_l(\theta^*, e^*)$.*

This theorem highlights the fact that the non-monotonicity of scaling laws is, in fact, something we should expect in large classes of games. Indeed, even under mild conditions one can show that there always exists a unilateral restriction that improves payoffs.

While the theorem guarantees the existence of a unilateral restriction, which improves equilibrium performance, we remark that it does not say anything about the ease with which one can find this restricted space. While the proof is constructive, it makes use of information of the environment's loss to construct the set—information that may not always be available to the learner. Furthermore, as we show in the following examples, the non-monotonicity can play out in complex ways.

**Example 1: Multi-Agent Reinforcement Learning**   We first demonstrate an extreme form of the reverse scaling predicted by Theorem 3.4 in the context of multi-agent reinforcement learning. To do so, we construct a Markov game in which the more the learner restricts their policy class, the more their expected payoff increases. Note that in keeping with the language of MARL, we consider the case when both players would like to maximize their long-run discounted rewards.

The Markov game in question is a two-player game with $n$ states. In each state $s_i$, with $i \in \{1, 2, \ldots, n\}$, both players have two actions available to them $\{0, 1\}$ with 0 corresponding to the top

---

[2]Our definition makes the assumption of strict improvement only on the learner's loss as that is what is important for future theorems.

row or left column. In each state, the environment is allowed to choose a policy that is unrestricted, meaning that $\pi_e(s) = [p, 1 - p]$ for any $p \in [0, 1]$. The learner can choose from policies such that: $\pi_l(s) = [p, 1 - p]$ for all $p \in [1 - \bar{p}, \bar{p}]$ for some $\bar{p} \in [0.5, 1]$. Varying $\bar{p}$ generates model classes of varying expressivity. For example, when $\bar{p} = 1$, then they are allowed to choose any policy, and for $\bar{p} = 0.5$, they are constrained to only playing uniform policies.

Given actions $a, b$ from the learner and environment, respectively, we define the transition probabilities as:
$$p(s_{i+1}|s_i, a = 0, b = 1) = p(s_{i+1}|s_i, a = 1, b = 1) = 1$$
$$p(s_i|s_i, a = 0, b = 0) = p(s_i|s_i, a = 1, b = 0) = 1$$

Thus, the transitions are deterministic, given the action of the environment. This results in the following utilities for the two players, which are simply their sum of discounted rewards[3]:
$$u_i(\pi_l, \pi_e) = \mathbb{E}_{\pi_l, \pi_e} \left[ \sum_{t=0}^{\infty} \gamma_i^t R_i(s_t, a_t, b_t) \right]$$

where $i \in \{e, l\}$ and $\gamma_e, \gamma_l$ are the player's discount factors. We construct the payoffs for the learner such that they have a dominant strategy of $\pi_l(s) = [\bar{p}, 1 - \bar{p}]$ in all states, and their expected cumulative payoff increases as the players end up further along the chain of states (as seen in Figure 1).

We construct the payoffs of the environment player such that they trigger a switch to the next state if and only if the probability that the learner puts on action 0 is below some threshold $p^*(s)$. We do so for a sequence of thresholds $p_i^*$ for $i = 1, ..., n$ such that $p_i^* > p_{i+1}^* > 0.5$. With this construction, for $p_{i+1}^* < \bar{p} < p_i^*$ the game will result in the players staying in state $s_i$ for all time. More details on the construction of the payoff matrices are left to Appendix E and a plot of the equilibrium rewards for the learner as a function of $\bar{p}$ is shown in Figure 1 for different discount factors for games having $n = 50$ states.

We empirically observe that as $\bar{p}$ decreases (i.e., the policy class of the learner is restricted), the performance of the learner behaves in non-monotonic ways and can, in fact, be made to increase as the policy class gets closer to the uniform policy. The highly non-convex nature of the case where $\gamma = 0.95$ also highlights the difficulty in choosing a model class in general since it can be posed as a non-convex optimization problem.

A key takeaway of this example is that in general-sum MARL, restrictive policy parametrizations like e.g., softmax policies or function approximation may not lead to worse performance at equilibrium like in competitive and single-agent RL [40]. Indeed our example suggests that the payoff in quantal response equilibria [41] of Markov games (i.e., equilibria in which agents constrain their strategies to a class of quantal responses–see e.g., [41]) can sometimes have a higher payoff than the unrestricted Nash equilibrium.

**Example 2: Participation dynamics**   Our second example is similar to problems considered in the literature on performative prediction [18] though the setup we consider also fits the literature on understanding participation dynamics [32, 21] and algorithmic collective action [9].

In this model, there is a base distribution $\mathcal{P}_0$ over the input-output space $\mathcal{X} \times \mathcal{Y}$ where $\mathcal{X}$ is feature space and $\mathcal{Y}$ is the output space. The learner is trying to perform supervised learning to learn a mapping $g : \mathcal{X} \mapsto \mathcal{Y}$. The environment, on the other hand, takes the form of a population of agents that selects a distribution on $\mathcal{P}$ on the input-output space (i.e., $\mathcal{P} \in \Delta(\mathcal{X}, \mathcal{Y})$) to maximize their own utility which depends on the choice of the learner.

The least restrictive class of models the learner has access to $\Omega$ is the set of all functions $g : \mathcal{X} \to \mathcal{Y}$. We also consider a restricted function class $\Theta$ which is the class of all functions $g_r : \mathcal{X}' \to \mathcal{Y}$ where $\mathcal{X}' \subset \mathcal{X}$ is the result of some feature mapping $\phi : \mathcal{X} \to \mathcal{X}'$. Thus, $\Theta$ is the space of all functions of the form $g_r(\phi(x))$. Clearly $\Theta \subset \Omega$.

We assume that the strategic manipulations of the environment take the form of manipulations to the data distribution which take place by mixing the base distribution $\mathcal{P}_0$ with a manipulated data distribution $\mathcal{P}_e$ such that the distribution seen by the learner is given by $\mathcal{P} = \alpha \mathcal{P}_e + (1 - \alpha)\mathcal{P}_0$ for some $\alpha \in [0, 1]$. The parameter $\alpha$ relates to the strength of the response distribution within the

---

[3]We switch from losses to utilities to emphasize the fact that players would like to maximize their reward.

mixture that the learner observes. It might represent the fraction of the population that engages in strategic manipulations of their data.

Finally, we assume that the learner is optimizing the zero-one loss, such that for any distribution $\mathcal{P}$, the best response $g$ or $g_r$ is the Bayes-Optimal classifier on $\mathcal{X}$ and $\mathcal{X}'$ respectively are:
$$g^*(x) = \arg\max_y \mathcal{P}(y|x) \ \& \ g_r^*(x) = \arg\max_y \mathcal{P}(y|\phi(x)).$$

Thus, the learner's loss at equilibrium is given by:
$$f_l(g^*, \mathcal{P}) = Pr(g^*(x) \neq y)$$
$$f_l(g_r^*, \mathcal{P}) = Pr(g_r^*(x) \neq y),$$

in each case, the probability is taken with respect to $\mathcal{P}$.

For the population of strategic agents, we assume that they would like the learner to avoid making use of certain 'protected' features and focus on a set of restricted features $\phi^*$. To do so, the strategic agents' response to the learner's model depends on the set of features it makes use of. Concretely, if the learner makes use of a set of features $\phi' : \mathcal{X} \to \mathcal{X}'$ that are more informative than some $\phi^* : \mathcal{X} \to \mathcal{X}^*$—i.e., $\mathcal{X}^* \subset \mathcal{X}'$, then the strategic agents add uniform noise to the base distribution, and if not they report their true data. This can be represented by the following utility function:

$$f_e(g, \mathcal{P}) = \begin{cases} TV(\mathcal{P}_e, U) : Pr(g(x) \neq g(\phi^*(x))) > 0 \\ TV(\mathcal{P}_e, \mathcal{P}_0) \text{ otherwise}, \end{cases}$$

where $U$ is the uniform distribution on $\mathcal{X} \times \mathcal{Y}$ and $TV$ represents the $TV$ distance between distributions.

As we will show, for sufficiently large $\alpha$, the learner is always better off optimizing over the less expressive model class at equilibrium. To do so, we assume that $\phi^*$ preserves enough information for the Bayes optimal classifier on the space $\mathcal{X}^*$ to be strictly better than random choice.

**Assumption 3.5.** Let $|\mathcal{Y}| = n$. Assume that the Bayes optimal classifier on $\mathcal{X}^*$ for $\mathcal{P}_0$ denoted $g_r^*(x) = \arg\max_{y \in \mathcal{Y}} \mathcal{P}(y|\phi^*(x))$ satisfies:

$$f_l(g_r^*, \mathcal{P}_0) = Pr(g_r^*(x) = y) < \frac{1}{n}$$

This leads to the following result for this game.

**Proposition 3.6.** *Under Assumption 3.5, consider two functions classes over which the learner can optimize, $\Omega$, and $\Theta$ which is the set of all functions from $g_r : \mathcal{X}^* \to \mathcal{Y}$, where $\mathcal{X}^* = \phi^*(\mathcal{X})$ such that $\phi^*(x) = x$ for $x \in \mathcal{X}^*$. Consider the corresponding games denoted $G$ and $G^*$, respectively. Then the Nash equilibrium in $G$ is $(g^*, \mathcal{P}^*)$ where $\mathcal{P}^* = (1 - \alpha)\mathcal{P}_0 + \alpha U$ and the Nash equilibrium in $G^*$ is given by $(g_r^*, \mathcal{P}_0)$. Furthermore, there exists a range of $\alpha \in (0, 1)$ such that:*
$$f_l(g^*, \mathcal{P}^*) > f_l(g_r^*, \mathcal{P}_0)$$

The proof of this proposition can be found in Appendix B.2. This proposition highlights the fact that interactions with strategic agents can make less expressive function classes yield better performance in strategic settings.

## 4  Online Learning for Model Selection in Games

The previous results emphasize the importance of careful model selection in strategic environments. In this section, we consider the problem of learning the best model class to optimize in strategic environments.

Due to the unknown and non-stationary nature of the environment, in game theoretic settings, the learner will have to interact repeatedly with the environment to learn which model class and, consequently, which strategy to play. Thus, we formulate a problem of learning in games in which the learner seeks to find the best model class across a set of candidate model classes as well as the best strategy. We frame this as a problem of *model selection* for games.

We remark that model-selection is an area of recent interest in online learning [42, 43], though—to the best of our knowledge—the paradigm has not been applied to games as yet. Most similar to

---

**Algorithm 1** Stochastic gradient descent to find Nash equilibrium in a strongly monotone game

---
1: **procedure** PSGD($\Theta, x_0, T$)
2:      **for** $t \leftarrow 1$ **to** $T$ **do**
3:          $\eta_t = \frac{2}{\mu(t+1)}$
4:          $x_{t+1} \leftarrow \Pi_{\Theta \times \mathcal{E}}(x_t - \eta_t \hat{F}(x_t))$
5:      **end for**
6:      $\hat{x}_T = $ **return** $\sum_{t=1}^{T} \frac{t}{T(T+1)/2} x_t$
7: **end procedure**

---

this problem is a line of work on meta-learning in games, which seeks to find good strategies that generalize across environments [44].

We describe an algorithm for how the learner can select model classes to identify which model class to use. As a proof-of-concept, we assume that all players use stochastic gradient descent and adopt the structure of a problem we analyzed in the Nash environment regime. In particular, we assume the learner has access to sets of subsets of $\Omega = \mathbb{R}^d$ and that their loss and the environment's loss satisfy the following generalization of Assumption 3.2. For simplicity, we let the tuple of a particular model and the environment action be denoted by $x$ (i.e., $(\theta, e) = x$). We note here that $F$ is the generalized gradient mapping as described in Definition 3.1.

**Assumption 4.1.** Assume the game defined on $f_e$ and $f_l$ is strongly monotone and that they are $L$-Lipschitz continuous on $\Omega \times \mathcal{E}$. Further, assume that the players have access to stochastic gradient estimators such that the estimated monotone mapping $\hat{F}$ satisfies, $\forall x \in \Omega \times \mathcal{E}$:

$$\mathbb{E}[\hat{F}(x)] = F(x) \text{ and } \mathbb{E}[\|\hat{F}(x) - F(x)\|^2] \leq \sigma^2.$$

Given this assumption and under the simplifying assumption that all players use decreasing stepsizes, we assume that for a given model class $\Theta_i$ the players engage in projected stochastic gradient descent of the form:

$$x_{t+1} = \Pi_{\Theta_i \times \mathcal{E}} \left( x_t - \eta_t \hat{F}(x_t) \right),$$

where $\Pi$ denotes the Euclidean projection onto $\Theta_i \times \mathcal{E}$. The pseudocode for this is described in Algorithm 1. We show that the running average of the iterates resulting from running this algorithm in an environment satisfying Assumption 4.1 concentrates quickly around the payoff at the Nash equilibrium. The proof of this proposition can be found in Appendix C.

**Proposition 4.2.** *Let $\Theta$ correspond to a particular model class which results in an instance of continuous action $\mu-$strongly monotone game with a unique Nash Equilibrium $x^*$. Under Assumption 4.1 and the assumption that all players use stepsize schedule $\eta_t = \frac{2}{\mu(t+1)}$, for any $\delta \in (0, 1)$ Algorithm 1 yields an estimate $\hat{x}_T$ such that:*

$$|f_l(\hat{x}_T) - f_l(x^*)| \leq \mathcal{O} \left( \frac{L^2 \log(\frac{1}{\delta}) + L^3}{\mu^2 T} \right),$$

*with probability at least $1 - \delta$.*

To derive this bound, we generalize an existing result from convex optimization [45]. Given these confidence bounds, we now propose a successive elimination algorithm for identifying the best model class in a game. The underlying assumption remains that the environment player is simply doing stochastic gradient descent. This should also extend to the case when the environment performs stochastic mirror descent [46]. The specific form of successive elimination is described in Algorithm 2.

As we show, this algorithm has strong properties in terms of identification of the best model class due to the fast concentration of our estimator from Proposition 4.2. We show the results with respect to identification and defer the proof to Appendix C.

**Proposition 4.3.** *Under the assumptions of Proposition 4.2, let $\mathcal{A} = \{\Theta_i\}_{i=1}^{n}$. With probability at least $1 - \delta$, Algorithm 2 identifies the model class whose Nash equilibrium yields the highest payoff after:*

$$\mathcal{O} \left( \frac{n(L^2 \log(\frac{n}{\delta}) + L^3)}{\mu^2 \Delta^*} \right)$$

---

**Algorithm 2** Successive Elimination for Best Design action Identification

---

1: **procedure** SUCCESSIVEELIMINATION($\{\Theta_i\}_{i=1}^n, \delta$)
2:     $S \leftarrow \{\Theta_i\}_{i=1}^n$
3:     $\tau = 1$
4:     **while** $|S| > 1$ **do**
5:         $T = \alpha 2^\tau$
6:         $\delta' \leftarrow \frac{\delta}{2nT^2}$
7:         **for all** $\Theta_i \in S$ **do**
8:             $x_T^i \leftarrow$ Algorithm 1 with $(\Theta_i, x_0, T)$
9:         **end for**
10:        $S \leftarrow S \setminus \{\Theta_i \in S : \exists \Theta_j$ such that:
    $f(x_T^i) + \frac{L^2 \log(\frac{1}{\delta'}) + L^3}{\mu^2 T} < f(x_T^j) - \frac{L^2 \log(\frac{1}{\delta'}) + L^3}{\mu^2 T}\}$
11:        $\tau = \tau + 1$
12:    **end while**
13:    **return** $S$
14: **end procedure**

---

*interactions with the environment, where $\Delta^*$ is the minimum suboptimality gap of the Nash equilibrium of a function class compared to that of the best function class.*

This result indicates that finding the best model class out of a set of candidate model classes may be computationally tractable in certain regimes. An interesting question that we leave for future work is whether it is possible to be no regret, not just within a model class, but across a set of model classes as well.

## 5    Conclusion

This work seeks to provide a framework for understanding the complexities that arise when models are released into strategic environments. We show that the prevailing understanding of scaling laws in machine learning fails to hold in large classes of strategic environments and show its implications for MARL and strategic classification, among other areas. Lastly we highlight a possible algorithmic solution to overcoming the problem of model selection in games in which we were able to design an algorithm to efficiently learn the best model class to optimize over without sacrificing performance in terms of regret.

Altogether, our results are a first step towards understanding scaling laws and hence, model selection in strategic environments. Our results suggest that we need to rethink our understanding of scaling laws before blindly deploying ever more complex models into real-world environments in which they will be faced with strategic behaviors. We leave many avenues of future work open, including questions about generalization and finite sample considerations, as well as the potential for more sophisticated algorithmic approaches to model selection.

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

# A  Learner-Environment Interactions

## A.1  Description of learner environment settings

We begin by providing a concrete description of each of the learner -environment settings we explore:

1. **Stationary Environments:** The environment has only one action (i.e., $\mathcal{E} = \{e\}$), and the problem reduces to that of classical ML. The resulting equilibrium is simply the minimum of the learner's loss given $e$ and the model class $\Theta_i$:

$$\theta^* = \arg\min_{\theta \in \Theta_i} f_l(\theta, e).$$

2. **Stackelberg Environments - Learner Leads:** The equilibrium outcome is the Stackelberg equilibrium of the two-player game under the assumption that the learner *leads*. This is, for example, the setup adopted in strategic classification [10]. The equilibrium is a joint strategy $(\theta^*, e^*)$ such that:

$$\theta^* = \arg\min_{\theta \in \Theta_i} f_l(\theta, BR_e(\theta)),$$

and $e^* = BR_e(\theta^*) = \arg\min_{e \in \mathcal{E}} f_e(\theta^*, e)$.

3. **Stackelberg Environments - Learner Follows:** The equilibrium outcome is the Stackelberg equilibrium of the two-player game under the assumption that the learner *follows*. This is, for example, the case that arises when agents attempt to perform data poisoning attacks [47] or engage in collective action [9] against the learner. Here the equilibrium is a joint strategy $(\theta^*, e^*)$ such that:

$$e^* = \arg\min_{e \in \mathcal{E}} f_e(BR_l(e), e),$$

and $\theta^* = BR_l(e^*) = \arg\min_{\theta \in \Theta} f_l(\theta, e^*)$.

4. **General Nash Environments:** Which allows us to model the general case when the interaction results in a Nash equilibrium. This is, for example, the desired solution in MARL [11] and participation and regression games [21, 32]. In this setting, the equilibrium outcome is a joint strategy $(\theta, e)$ such that:

$$f_e(\theta, e') \geq f_e(\theta, e) \ \forall \, e' \in \mathcal{E},$$
$$f_l(\theta', e) \geq f_l(\theta, e) \ \forall \, \theta' \in \Theta_i.$$

We remark that in this last case, the assumption of a two-player game is made for simplicity, and our results would go through in $n$-player games.

## A.2  Stationary environments and Stackelberg games with the learner leading

We investigate the two cases in which performance monotonically increases as a function of complexity: stationary environments and Stackelberg interactions where the learner has commitment power (i.e., the learner "leads"). This fact follows from the elementary observation that in both of these regimes, the learner simply solves the same optimization problem over a larger space.

**Proposition A.1.** *Let $\mathbb{A}$ be an ordered set. Consider two model classes, $\Theta_i, \Theta_j \in \mathbb{A}$ with $i < j$. If $(\theta_i, e_i)$ and $(\theta_j, e_j)$ are equilibrium outcomes in stationary environments or Stackelberg environments in which the learner leads with the instantiated model classes being $\Theta_i$ and $\Theta_j$ respectively, then $f_l(\theta_i, e_i) \geq f_l(\theta_j, e_j)$*

*Proof.* For a stationary game, the proposition above follows naturally. We know that $e_i = e_j$ which we denote $e^*$. Since $\theta_i \in \Theta_i \subseteq \Theta_j$, the equilibrium $(\theta_j, e^*)$ necessarily must be such that $f_l(\theta_j, e^*) \leq f_l(\theta_i, e^*)$, otherwise $(\theta_j, e^*)$ is not an equilibrium point.

For a Stackelberg game where the learner leads, the proposition follows the same argument. According to Stackelberg dynamics, we know that $BR(\theta_i) = e_i$ and $BR(\theta_j) = e_j$. We can see that it must be the case that $f_l(\theta_j, e_j) \leq f_l(\theta_i, e_i)$ otherwise simply selecting $\theta_i$ when optimizing in $\Theta_j$ would be a profitable deviation. $\qquad\square$

While this proposition is trivial to prove, it has implications for adversarial machine learning and strategic classification. Adversarial learning can be modeled as zero-sum or min-max games [7]. In these games, if the Nash equilibrium exists, it coincides with the Stackelberg equilibria via simple min-max theorems [48]. This implies that the basic intuition of scaling laws holds true for adversarial learning. Similar takeaways hold true for strategic classification because it is commonly modeled as a Stackelberg game in which the learner leads.

### A.3 Stackelberg games where the learner follows

We also consider the case in which the environment results in a Stackelberg equilibrium where the learner follows. In lieu of a general result for this case, we construct a simple example in strategic linear regression that highlights the fact that when the learner follows in a Stackelberg game—as happens in settings such as collective action and non-adversarial backdoor attacks in machine learning— the use of more features can actually hurt.

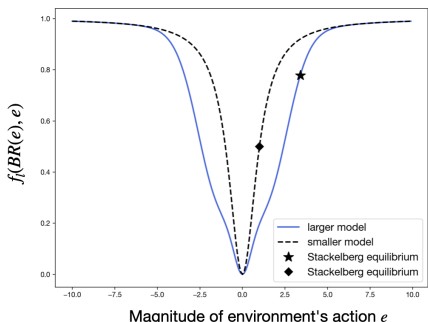

Figure 2: The loss for the learner at their best response in a regression game as the magnitude of the environment's perturbation vector varies with the payoffs achieved at equilibrium as derived in Proposition A.2.

**Example 3: Strategic Linear Regression** In this setup, we have a decision maker who would like to solve a regression problem and is choosing between different feature mappings they can use to fit a model. More precisely, we consider a learner deciding between whether to pick $\phi_\theta^1(x) = \theta^T x$ or $\phi_\theta^2(x) = \theta_1^T x + \theta_2 \exp(-\|x\|^2)$ as the function to use for the regression task. The classifier's goal is to learn $\theta$ so as to minimize the expected squared error. In this framework, the environment can add a deviation $e$ to the dataset. This would mean that the input to the regressor model would be $x + e$.

Consider the model classes $\Theta_{\phi_\theta^1(x)}, \Theta_{\phi_\theta^2(x)}$ derived from $\phi_\theta^1(x), \phi_\theta^2(x)$ respectively. We show that despite $\Theta_{\phi_\theta^1(x)} \subset \Theta_{\phi_\theta^2(x)}$, at a Stackelberg equilibrium, the learner has a higher payoff when they learn from $\Theta_{\phi_\theta^1(x)}$ as opposed to $\Theta_{\phi_\theta^2(x)}$. This is in stark contrast to what happens in stationary environments in which adding features never hurts performance since they can just be given a weight of 0. More concretely:

**Proposition A.2.** *Consider a dataset where each data point $x \in \mathbb{R}^d$ is drawn from a distribution $\mathcal{D}$. A learner has the option to select one of two model classes:*

$$\Theta_{\phi_\theta^1(x)} = \{\phi_\theta^1(x) : \theta \in \mathbb{R}^d \text{ where: } \phi_\theta^1(x) = \theta^T x\}$$
$$\Theta_{\phi_\theta^2(x)} = \{\phi_\theta^2(x) : \theta_1, \theta_2 \in \mathbb{R}^d \times \mathbb{R}$$
$$\text{where: } \phi_\theta^2(x) = \theta_1^T x + \theta_2 \exp(-\|x\|^2)\}.$$

*Assume that each data point can be perturbed by an error vector $e \in C \subseteq \mathbb{R}^d$, resulting in the learner observing $x + e$ instead of $x$. For some compact convex $C$, distribution $\mathcal{D}$ and dimension $d$, the Stackelberg equilibrium attained by optimizing over $\Theta_{\phi_\theta^1(x)}$ results in a strictly lower loss than that attained by optimizing over $\Theta_{\phi_\theta^2(x)}$.*

The calculations for this proposition are in the Appendix D. Figure 2 highlights the non-monotonicity of performance between different equilibria in the two spaces. We see that at the Stackelberg

equilibrium, the larger model class incurs a greater loss than the smaller model class despite the fact that the loss incurred by the larger model class is lower than that incurred by, the lower model class in a point-wise sense.

# B    Additional proofs

## B.1    Proof of Theorem 3.4

We begin by revisiting the main assumptions of this proof:

**Assumption B.1** (Restatement of Assumption 3.2). Assume the game defined on $f_e$ and $f_l$ is strongly monotone on $\Omega \times \mathcal{E}$. Further assume that

1. $f_l$ and $f_e$ are jointly convex in $\theta$ and $e$.

2. The gradient mappings, $\nabla f_l$ and $\nabla f_e$ exist and are well defined for all $(\theta, e)$. Furthermore, the gradient mappings are $L$-Lipschitz continuous in the joint action space.

3. The Nash equilibrium $\theta^* \in \Theta$ is on the interior of $\Theta$ with $\nabla_\theta BR_e(\theta^*) \neq 0$.

**Theorem B.2** (Restatement of Theorem 3.4). *For a two-player monotone game $G$ on $\Theta \times \mathcal{E}$ which satisfies Assumption 3.2, if the unique Nash equilibrium in $\Theta \times \mathcal{E}$ is* not *Pareto optimal then there exists a restriction of the learner's model class (i.e., a set $\Theta' \subset \Theta$) such that the restricted game $G'$ on $\Theta' \times \mathcal{E}$ admits a Nash equilibrium $(\theta', e')$ with: $f_l(\theta', e') < f_l(\theta^*, e^*)$.*

*Proof of Theorem 3.4.* Note the following: because we can achieve strictly lower loss for the jointly convex loss function $f_l$, we know that $\nabla f_l(\theta^*, e^*) \neq 0$

Consider the following the function $\bar{f}_l(\theta) := f_l(\theta, BR_e(\theta))$. Here $BR_e(\theta)$ is the best response of the other player to the action $\theta$, i.e., $BR_e(\theta) := \arg\min_{e \in \mathcal{E}} f_e(\theta, e)$.

Relying on this function definition, we describe a particular restriction on the model class $\Theta$ and then find a Nash equilibrium for this model class, which is different from $(\theta^*, e^*)$ and whose loss for the $\Theta-$player is lower in this new equilibrium.

Realize that $\nabla \bar{f}_l(\theta^*) = \nabla_\theta f_l(\theta^*, BR_e(\theta^*)) + \nabla_\theta BR_e(\theta^*) \nabla_e f_l(\theta^*, BR_e(\theta^*)) \neq 0$. In particular, we note that $\nabla_\theta BR_e(\theta^*) \cdot \nabla_e f_l(\theta^*, BR_e(\theta^*)) \neq 0$ since if it were, it would mean that either $\nabla_\theta BR_e(\theta^*)$ or $\nabla_e f_l(\theta^*, BR_e(\theta^*))$ were equal to zero which is not the case. $\nabla_e f_l(\theta^*, BR_e(\theta^*)) = 0$, would mean that both $\nabla_\theta f_l(\theta^*, BR_e(\theta^*))$ and $\nabla_e f_l(\theta^*, BR_e(\theta^*)) = 0$ implying a global minimum which would contradict the existence of a Pareto improving point. $\nabla_\theta BR_e(\theta^*) \neq 0$ follows from the Assumption 3.2.

Noting the fact that $\nabla \bar{f}_l(\theta^*) \neq 0$ allows us to pick a direction with respect to the inner product with $\nabla \bar{f}_l(\theta^*)$, let $v$ be any vector $\in \mathbb{R}^{d_\theta}$ such that $\langle v, \nabla \bar{f}_l(\theta^*) \rangle > 0$. Consider $\theta' = \theta^* - \delta v$ for some $\delta > 0$. Let $e' = BR_e(\theta')$. We will now define $\Theta' \subset \Theta$ such that the game on the model class $\Theta' \times \mathcal{E}$ has $(\theta', e')$ as a Nash equilibrium

Let $\tilde{\Theta} = \{\theta \in \Theta : \langle \theta - \theta', v \rangle \leq 0\}$. We then go on to define $\Theta' = \{\theta \in \tilde{\Theta} : \langle \nabla_\theta f_l(\theta', BR_e(\theta')), \theta' - \theta \rangle \leq 0\}$. Notice how the first step removes $(\theta^*, e^*)$ from the construction. The second step makes it such that $BR_\theta(e') = \theta'$ for all $\theta \in \Theta'$. Since $e'$ is the best response to $\theta'$ we get that the point $(\theta', e')$ is a Nash equilibrium.

What is left to show is that we can create such a restriction with the characteristic that the $\Theta$-player's loss function is lowered at the new equilibrium. To do this, we rely on the choice of the $\delta$ parameter.

**Claim B.3.** *There exists a value of $\delta > 0$ such that $f_l(\theta', e') < f_l(\theta^*, e^*)$*

To see this, consider the Taylor expansion of $\bar{f}_l(\theta')$ at $\theta^*$.

$$\bar{f}_l(\theta') = \bar{f}_l(\theta^*) + \langle \nabla \bar{f}_l(\theta^*), \theta' - \theta^* \rangle + \frac{1}{2}(\theta' - \theta^*)^T H_\theta(\bar{\theta})(\theta' - \theta^*)$$

$$\leq \bar{f}_l(\theta^*) + \langle \nabla \bar{f}_l(\theta^*), \theta' - \theta^* \rangle + \frac{L}{2}\|\theta' - \theta^*\|^2$$

$$= \bar{f}_l(\theta^*) - \delta\langle \nabla \bar{f}_l(\theta^*), v \rangle + \delta^2 \frac{L}{2}\|v\|^2$$

In the first line $\bar{\theta} \in \{\theta \in \Theta : \theta = \lambda\theta^* + (1-\lambda)\theta', \lambda \in [0,1]\}$. We note that by our construction $\langle \nabla \bar{f}_l(\theta^*), v \rangle > 0$. Furthermore, we realize that the term $\delta \langle \nabla \bar{f}_l(\theta^*), v \rangle \in \mathcal{O}(\delta)$ and that $\delta^2 \frac{L}{2}\|v\|^2 \in \mathcal{O}(\delta^2)$. This means we can select a value of $\delta$ such that the term $\delta \langle \nabla \bar{f}_l(\theta^*), v \rangle - \delta^2 \frac{L}{2}\|v\|^2$ is positive. With this, we can then deduce that $\bar{f}_l(\theta^*) > \bar{f}_l(\theta')$ which then completes the proof. $\qquad\square$

### B.2 Other proofs in Section 3

We begin with proving the proposition on Strategic classification. Here we restate the assumptions again:

**Assumption B.4** (Restatement of Assumption 3.5). Let $|\mathcal{Y}| = n$. Assume that the Bayes optimal classifier on $\mathcal{X}^*$ for $\mathcal{P}_0$ denoted $g_r^*(x) = \arg\max_{y \in \mathcal{Y}} \mathcal{P}(y|\phi^*(x))$ satisfies:

$$f_l(g_r^*, \mathcal{P}_0) = Pr(g_r^*(x) = y) < \frac{1}{n}$$

**Proposition B.5** (Restatement of Proposition 3.6). *Under Assumption 3.5, consider two functions classes over which the learner can optimize, $\Omega$, and $\Theta$ which is the set of all functions from $g_r : \mathcal{X}^* \to \mathcal{Y}$, where $\mathcal{X}^* = \phi^*(\mathcal{X})$ such that $\phi^*(x) = x$ for $x \in \mathcal{X}^*$. Consider the corresponding games denoted $G$ and $G^*$ respectively. Then the Nash equilibrium in $G$ is $(g^*, \mathcal{P}^*)$ where $\mathcal{P}^* = (1-\alpha)\mathcal{P}_0 + \alpha U$ and the Nash equilibrium in $G^*$ is given by $(g_r^*, \mathcal{P}_0)$. Furthermore, there exists a range of $\alpha \in (0,1)$ such that:*

$$f_l(g^*, \mathcal{P}^*) > f_l(g_r^*, \mathcal{P}_0)$$

*Proof of Proposition 3.6.* To prove the first part of this proposition, we note that $(g^*, \mathcal{P}^*)$ is a Nash equilibrium in that no player has any incentive to unilaterally deviate given their action sets. Indeed, if $g^*$ is the Bayes-optimal classifier on $P^*$ in $\Omega$ then $Pr(g(x) \neq g(\phi^*(x))) > 0$ and consequently the environment's best-response perturbation is $\mathcal{P}_e = U$. Similarly, $g_r^*, \mathcal{P}_0$ is the Bayes-optimal classifier on $\mathcal{X}^*$ and satisfies $Pr(g(x) = g(\phi^*(x))) = 0$ by definition. As such, the environment's best response is given by $\mathcal{P}_e = \mathcal{P}_0$.

To prove the second part of the proof we note that:

$$f_l(g^*, \mathcal{P}^*) \geq (1-\alpha) \min_{g \in \Omega} f_l(g, \mathcal{P}_0) + \frac{\alpha}{n} > \frac{\alpha}{n}$$

Now, note that for $1 \geq \alpha \geq n f_l(g_r^*, \mathcal{P}_0)$, we have that:

$$f_l(g^*, \mathcal{P}^*) > f_l(g_r^*, \mathcal{P}_0)$$

$\qquad\square$

## C Further results on Online Learning for Model Selection in Games

We now present a proof for convergence for the stochastic gradient descent algorithm. We restate the main assumptions here as well:

**Assumption C.1** (Restatement of Assumption 4.1). Assume the game defined on $f_e$ and $f_l$ is strongly monotone on $\Omega \times \mathcal{E}$. Further, assume that

1. The functions $f_l$ and $f_e$ are $L$-Lipschitz continuous on $\Omega \times \mathcal{E}$.

2. The players have access to stochastic gradient estimators such that the estimated monotone mapping $\hat{F}$ satisfies, $\forall x \in \Omega \times \mathcal{E}$:

$$\mathbb{E}[\hat{F}(x)] = F(x) \quad \text{and} \quad \mathbb{E}[\|\hat{F}(x) - F(x)\|^2] \leq \sigma^2.$$

**Proposition C.2** (Restatement of Proposition 4.2). *Let $\Theta$ correspond to a particular model class which results in an instance of continuous action $\mu-$strongly monotone game with a unique Nash Equilibrium $(x^*)$. Under Assumption 4.1 and the assumption that all players use stepsize schedule $\eta_t = \frac{2}{\mu(t+1)}$, for any $\delta \in (0,1)$ Algorithm 2 yields an estimate $\hat{x}_T$ such that:*

$$|f_l(\bar{x}) - f_l(x^*)| \leq \mathcal{O}\left(\frac{L^2 \log(\frac{1}{\delta}) + L^3}{\mu^2 T}\right),$$

*with probability at least $1 - \delta$.*

*Proof of Proposition 4.2.* Let $x_t = (x_1, \ldots, x_n)$ for a fixed number of players. We then define

$$F(x_t) = \begin{bmatrix} \nabla_1 u_1(x_t) \\ \nabla_2 u_2(x_t) \\ \vdots \\ \nabla_n u_n(x_t) \end{bmatrix}. \text{ Let } \hat{F}(x_t) = F(x_t) - \hat{E}(x_t), \text{ where } \mathbb{E}[\hat{E}(x_t)] = \mathbf{0} \text{ and } \|\hat{E}(x_t)\|_2 \le 1 \text{ a.s..}$$

From the description of Algorithm 2, we can see that $x_{t+1} = \Pi_{\Theta \times \mathcal{E}}(x_t - \eta_t \hat{F}(x_t))$ The proof of this proposition closely mirrors [45]. Let $\bar{x}$ be the output of Algorithm 1. We can see that:

$$
\begin{aligned}
\|x_{t+1} - x^*\|^2 &= \|\Pi_{\Theta \times \mathcal{E}}(x_t - \eta_t \hat{F}(x_t)) - x^*\|^2 \\
&\le \|x_t - \eta_t \hat{F}(x_t) - x^*\|^2 \\
&= \|x_t - x^*\|^2 - 2\eta_t \langle \hat{F}(x_t), x_t - x^* \rangle + \eta_t^2 \|\hat{F}(x_t)\|^2 \\
&= \|x_t - x^*\|^2 - 2\eta_t \langle \hat{F}(x_t), x_t - x^* \rangle - 2\eta_t \langle F(x_t), x_t - x^* \rangle + 2\eta_t \langle F(x_t), x_t - x^* \rangle + \eta_t^2 \|\hat{F}(x_t)\|^2 \\
&= \|x_t - x^*\|^2 + 2\eta_t \langle F(x_t) - \hat{F}(x_t), x_t - x^* \rangle - 2\eta_t \langle F(x_t), x_t - x^* \rangle + \eta_t^2 \|\hat{F}(x_t)\|^2
\end{aligned}
$$

Rearranging terms we get:

$$
\begin{aligned}
2\eta_t \langle F(x_t), x_t - x^* \rangle &\le \|x_t - x^*\|^2 - \|x_{t+1} - x^*\|^2 + 2\eta_t \langle F(x_t) - \hat{F}(x_t), x_t - x^* \rangle + \eta_t^2 \|\hat{F}(x_t)\|^2 \\
\langle F(x_t), x_t - x^* \rangle &\le \frac{\|x_t - x^*\|^2 - \|x_{t+1} - x^*\|^2}{2\eta_t} + \langle \hat{E}(x_t), x_t - x^* \rangle + \frac{\eta_t}{2} \|\hat{F}(x_t)\|^2 \\
\frac{1}{2}\langle F(x_t), x_t - x^* \rangle &\le \frac{\|x_t - x^*\|^2 - \|x_{t+1} - x^*\|^2}{2\eta_t} - \frac{1}{2}\langle F(x_t), x_t - x^* \rangle + \langle \hat{E}(x_t), x_t - x^* \rangle + \frac{\eta_t}{2} \|\hat{F}(x_t)\|^2 \\
\langle F(x_t), x_t - x^* \rangle &\le 2 \cdot \left( \frac{\|x_t - x^*\|^2 - \|x_{t+1} - x^*\|^2}{2\eta_t} - \frac{\mu}{2}\|x_t - x^*\|^2 + \langle \hat{E}(x_t), x_t - x^* \rangle + \frac{\eta_t}{2} \|\hat{F}(x_t)\|^2 \right) \\
t\langle F(x_t), x_t - x^* \rangle &\le 2t \cdot \left( \frac{\|x_t - x^*\|^2 - \|x_{t+1} - x^*\|^2}{2\eta_t} - \frac{\mu}{2}\|x_t - x^*\|^2 + \langle \hat{E}(x_t), x_t - x^* \rangle + \frac{\eta_t}{2} \|\hat{F}(x_t)\|^2 \right) \\
&= 2 \cdot \left( t\left(\frac{1}{2\eta_t} - \frac{\mu}{2}\right)\|x_t - x^*\|^2 - \frac{t}{2\eta_t}\|x_{t+1} - x^*\|^2 + t\langle \hat{E}(x_t), x_t - x^* \rangle + t\frac{\eta_t}{2} \|\hat{F}(x_t)\|^2 \right) \\
&= 2 \cdot \left( \frac{\mu}{4} \cdot \left(t(t-1)\|x_t - x^*\|^2 - t(t+1)\|x_{t+1} - x^*\|^2\right) + t\langle \hat{E}(x_t), x_t - x^* \rangle + \frac{t}{\mu(t+1)}\|\hat{F}(x_t)\|^2 \right) \\
&\le 2 \cdot \left( \frac{\mu}{4} \cdot \left(t(t-1)\|x_t - x^*\|^2 - t(t+1)\|x_{t+1} - x^*\|^2\right) + t\langle \hat{E}(x_t), x_t - x^* \rangle + \frac{(L+1)^2}{\mu} \right)
\end{aligned}
$$

We then sum over all $t$ and noting that the right-hand side has a portion with a telescoping sum, we see that:

$$
\begin{aligned}
\sum_{t=1}^{T} t\langle F(x_t), x_t - x^* \rangle &\le 2 \cdot \left( \sum_{t=1}^{T} t\langle \hat{E}(x_t), x_t - x^* \rangle + \frac{T \cdot (L+1)^2}{\mu} \right) \\
\frac{1}{T(T+1)/2}\sum_{t=1}^{T} t\langle F(x_t), x_t - x^* \rangle &\le \frac{4}{T(T+1)} \cdot \left( \sum_{t=1}^{T} t\langle \hat{E}(x_t), x_t - x^* \rangle + \frac{T \cdot (L+1)^2}{\mu} \right)
\end{aligned}
$$

By monotonicity we get note that: $\mu\|x_t - x^*\|^2 \leq \langle F(x_t), x_t - x^*\rangle$ and thus:

$$\frac{\mu}{T(T+1)/2}\sum_{t=1}^{T} t\|x_t - x^*\|^2 \leq \frac{4}{T(T+1)}\cdot\left(\sum_{t=1}^{T} t\langle\hat{E}(x_t), x_t - x^*\rangle + \frac{T\cdot(L+1)^2}{\mu}\right)$$

$$\frac{1}{T(T+1)/2}\sum_{t=1}^{T} t\|x_t - x^*\|^2 \leq \frac{4}{\mu T(T+1)}\cdot\left(\sum_{t=1}^{T} t\langle\hat{E}(x_t), x_t - x^*\rangle + \frac{T\cdot(L+1)^2}{\mu}\right)$$

$$\|\sum_{t=1}^{T}\frac{t}{T(T+1)/2}x_t - x^*\|^2 \leq \frac{4}{\mu T(T+1)}\cdot\left(\sum_{t=1}^{T} t\langle\hat{E}(x_t), x_t - x^*\rangle + \frac{T\cdot(L+1)^2}{\mu}\right)$$

$$\|\bar{x} - x^*\|^2 \leq \frac{4}{\mu T(T+1)}\cdot\left(\sum_{t=1}^{T} t\langle\hat{E}(x_t), x_t - x^*\rangle + \frac{T\cdot(L+1)^2}{\mu}\right)$$

$$|f_i(\bar{x}) - f_i(x^*)| \leq \frac{4L}{\mu T(T+1)}\cdot\left(\underbrace{\sum_{t=1}^{T} t\langle\hat{E}(x_t), x_t - x^*\rangle}_{E_T} + \frac{T\cdot(L+1)^2}{\mu}\right)$$

To complete the proof, we rely on a high probability bound on $E_T$ which makes use of a specialized form of the Generalized Freedman's Inequality.

**Lemma C.3** ([45] Lemma 4.1). *Let* $E_T = \sum_{t=1}^{T} t\langle\hat{E}(x_t), x_t - x^*\rangle$. *Then for any* $\delta \in (0,1)$ *we have that* $E_T \leq \mathcal{O}\left(\frac{L}{\mu}\cdot T\log(\frac{1}{\delta})\right)$

plugging in the bound on $E_T$ completes the proof. $\qquad\square$

We then proceed to provide a proof of the successive elimination protocol. This proof which closely follows [49]

**Proposition C.4** (Restatement of Proposition 4.3). *Under the assumptions of Proposition 4.2, let* $\mathcal{A} = \{\Theta_i\}_{i=1}^{n}$. *With probability at least* $1 - \delta$, *Algorithm 2 identifies the model class whose Nash equilibrium yields the highest payoff after:*

$$\mathcal{O}\left(\frac{n(L^2\log(\frac{n}{\delta}) + L^3)}{\mu^2\Delta^*}\right)$$

*interactions with the environment, where* $\Delta^*$ *is the minimum suboptimality gap of the Nash equilibrium of a function class compared to that of the best function class.*

*Proof of 4.3.* We begin by showing an "anytime" confidence interval bound.

**Lemma C.5.** *Let* $X_{i,T} = f(x_T^i)$ *where* $x_T^i \leftarrow$ *Algorithm* $1(\Theta_i, x_0, T)$ *and* $X_i^* = f(x_i^*)$ *where* $x_i^*$ *is the Nash equilibrium point for* $\Theta_i$. *We then have that:* $\mathbb{P}\left(\bigcup_{i=1}^{n}\left\{\bigcup_{T=1}^{\infty}\left\{|X_{i,T} - X_i^*| \geq \frac{L^2\log(\frac{2T^2n}{\delta}) + L^3}{\mu^2 T}\right\}\right\}\right) \leq \delta$

*Proof.* Here we rely on the union bound to note that:

$$\mathbb{P}\left(\bigcup_{i=1}^{n}\left\{\bigcup_{T=1}^{\infty}\left\{|X_{i,T} - X_i^*| \geq \frac{L^2\log(2\frac{T^2n}{\delta}) + L^3}{\mu^2 T}\right\}\right\}\right) \leq \sum_{i=1}^{n}\sum_{T=1}^{\infty}\mathbb{P}\left(|X_{i,T} - X_i^*| \geq \frac{L^2\log(\frac{2T^2n}{\delta}) + L^3}{\mu^2 T}\right)$$

$$\leq \sum_{i=1}^{n}\frac{\delta}{2n}\sum_{T=1}^{\infty}\frac{1}{T^2}$$

$$\leq \delta$$

$\qquad\square$

**Lemma C.6.** *With probability greater than or equal to $1 - \delta$, the best model class $\Theta_k$, is retained in the active set $S$ until the end of Algorithm 2.*

*Proof.* Let $\mathcal{D}$ be the event $\bigcup\limits_{i=1}^{n} \left\{ \bigcup\limits_{T=1}^{\infty} \left\{ |X_{i,T} - X_i^*| \geq \frac{L^2 \log(2\frac{T^2 n}{\delta}) + L^3}{\mu^2 T} \right\} \right\}$. We know that $\mathcal{D}^C$ occurs with probability at least $1 - \delta$. Let $U(T, \delta) = \frac{L^2 \log(2\frac{T^2 n}{\delta}) + L^3}{\mu^2 T}$, $\Theta_k$ is dropped if there exists $j, T$ such that $X_{j,T} - U(T, \delta) > X_{k,T} + U(T, \delta)$. We consider the scenario where the event $\mathcal{D}^C$ occurs. In this scenario we have that $X_j^* + U(T, \delta) \geq X_{j,T}$ and that $X_{k,T} \geq X_k^* - U(T, \delta)$ for all $T$. Plugging these two inequalities into the first expression gives us that $X_j^* \geq X_k^*$ which is a contradiction. Therefore, with probability greater than or equal to $1 - \delta$ we have the best model class $\Theta_i$ remaining in $S$. $\qquad\square$

**Lemma C.7.** *Given that the best model class $\Theta_k$ is identified by Algorithm 2, it will terminate after $\mathcal{O}\left( \frac{n(L^2 \log(\frac{n}{\delta}) + L^3)}{\mu^2 \Delta^*} \right)$ samples*

*Proof.* Let $\Delta_i = X_k^* - X_i^*$. Let $\Delta^* = \min_i \Delta_i$. Let $\Theta_k$ be the action that corresponds to the highest payoff at the Nash equilibrium point.
We note that one of the conditions which leads to action $\Theta_i$ being removed from the consideration set $S$ is

$$X_{k,T} - U(T, \delta) \geq X_{i,T} + U(T, \delta) \tag{1}$$

Assuming that the event $\mathcal{D}^C$ holds, for each model class $\Theta_i$ we have that, $X_{k,T} \geq X_k^* - U(t, \delta)$ and that $X_{i,t} \leq X_i^* + U(t, \delta)$. Substituting these expressions into what we have by 1, we get that :

$$X_k^* - X_i^* \geq 2U(T, \delta) + 2U(T, \delta)$$
$$\Delta_i \geq 4U(T, \delta)$$

Now we consider the case where $\Delta_i = \Delta^*$ to find a bound for $T$.

$$\Delta^* \geq 4U(T, \delta)$$
$$\Delta^* \geq \frac{4(L^2 \log(\frac{2T^2 n}{\delta}) + L^3)}{\mu^2 T}$$
$$\mu^2 T - \frac{8L^2 \log(T)}{\Delta^*} \geq \frac{4(L^2 \log(\frac{2n}{\delta}) + L^3)}{\Delta^*}$$
$$\mu^2 T \geq \frac{4(L^2 \log(\frac{2n}{\delta}) + L^3)}{\Delta^*}$$
$$T \geq \mathcal{O}\left( \frac{L^2 \log(\frac{n}{\delta}) + L^3}{\mu^2 \Delta^*} \right)$$

From here we proceed to find the $\tau$ which corresponds to $\mathcal{O}\left( \frac{L^2 \log(\frac{n}{\delta}) + L^3}{\mu^2 \Delta^*} \right)$ steps which is simply found by taking the log. We then note that $\sum\limits_{\tau=1}^{\mathcal{O}\left( \log(\frac{L^2 \log(\frac{n}{\delta}) + L^3}{\mu^2 \Delta^*}) \right)} 2^\tau = \mathcal{O}\left( \frac{L^2 \log(\frac{n}{\delta}) + L^3}{\mu^2 \Delta^*} \right)$. Summing over $n - 1$ decision actions we get $\mathcal{O}\left( \frac{n(L^2 \log(\frac{n}{\delta}) + L^3)}{\mu^2 \Delta^*} \right)$ samples which completes the proof. $\qquad\square$

$\square$

# D   Additional calculations for Linear Regression Example

Consider the following setup. The decision-maker would like to solve a regression problem and has the choice of two different regression models. For a distribution of input data $\mathcal{D}$, and a datapoint $x$, they can either compute:

$$\phi_\theta^1(x) = \theta^T(x + e)$$

Or:
$$\phi_\theta^2(x) = \theta_1^T(x+e) + \theta_2 \exp(-\|x+e\|^2)$$

For the rest of the calculation we take the distribution of the input data to be $\mathcal{N}(0, I)$. Suppose the true relationship between features $x$ and outcomes $y$ is linear and is by $y = \beta^T x$. However, the training data is reported by a population of strategic agents who commit to all manipulating their features in the same way such that the reported features are given by $x' = x + e$. Equivalently, this can be seen as the input data being misreported and generated from a distribution $\mathcal{N}(e, I)$.

Suppose the population of strategic agents knows that the learner will solve a regression problem. Then, they would like to choose $e$ to maximize their expected prediction given. Concretely:
$$e^* = \arg\max_e \mathbb{E}[\phi_{\theta^*}^i(x+e)]$$

Given the fact that $\theta^* = \arg\min_\theta \mathbb{E}[(y - \phi_\theta(x+e))^2]$. For this example, the set $C = \{e \in \mathbb{R}^d : e = k\frac{\beta}{\|\beta\|}$ for $k \in [-10, 10]\}$. We select this direction because, at a high level, the best deviation for the environment can be shown for many model classes to be in the direction of $\beta$. As for the magnitude boundaries, the equilibrium points we found lie in the interior of the set, and hence, there was nothing special about the boundaries selected for this example.

**Case 1: Small model** For this model, we do not rely explicitly on the definition of $C$. We find that the Stackelberg equilibrium action for the environment over $\mathbb{R}^d$ already lies in $C$. As such, this calculation does not make use of the structure of $C$.

To begin, we compute the the optimal $\theta$ for a given $e$ when $\phi_\theta(x) = \theta^T(x+e)$:
$$\begin{aligned}
\nabla_\theta f_l(e, \theta) &= \nabla_\theta \mathbb{E}[(y - \phi_\theta(x))^2] \\
&= \nabla_\theta \mathbb{E}[(\beta^T x - \theta^T(x+e))^2] \\
&= 2\beta - 2\theta - 2ee^T\theta
\end{aligned}$$

Setting this equal to 0, we find that:
$$\theta^*(e) = \left(I - \frac{ee^T}{1 + \|e\|^2}\right)\beta$$

plugging this into the problem for the strategic agents, we find that:
$$\begin{aligned}
e^* &= \arg\max_e \mathbb{E}[\phi_{\theta^*}(x+e)] \\
&= \arg\max_e \theta^*(e)^T e \\
&= \arg\max_e \frac{e^T\beta}{1 + \|e\|^2}
\end{aligned}$$

This results in the optimal choice of $e$ for the population of strategic agents being $e^* = \frac{\beta}{\|\beta\|}$, which in turn results in the regression accuracy of the decision-maker being:
$$f_l(e^*, \theta^*(e^*)) = \frac{1}{2}\|\beta\|^2$$

**Case 2: Larger model** In this case, while we make use of the structure of $C$, numerical experiments suggest that this point may be an equilibrium point over a far larger set than $C$. As this was an illustrative example, we did not venture to formally prove that the point we found was a Stackelberg equilibrium point across $\mathbb{R}^d$. For the second case, let us first expand the loss for the decision-maker as:
$$f_l(e, \theta) = \mathbb{E}[(\beta^T x - \theta_1^T(x+e))^2] - 2\theta_2\mathbb{E}[\exp(-\|x+e\|^2)(\beta^T x - \theta_1^T(x+e))] + \theta_2^2\mathbb{E}[\exp(-2\|x+e\|^2)]$$

We first evaluate $-2\theta_2\mathbb{E}[\exp(-\|x+e\|^2)(\beta^T x - \theta_1^T(x+e))]$. Note that $-2\theta_2\mathbb{E}[\exp(-\|x+e\|^2)(\beta^T x - \theta_1^T(x+e))] = -2\theta_2\beta^T\mathbb{E}[\exp(-\|x+e\|^2)x] + 2\theta_2\theta_1^T\mathbb{E}[\exp(-\|x+e\|^2)x] +$

$2\theta_2\theta_1^T\mathbb{E}[\exp(-\|x+e\|^2)e]$

$$\mathbb{E}[\exp(-\|x+e\|^2)x] = (\frac{1}{2\pi})^{\frac{d}{2}}\int_{\mathbb{R}^d} x\exp(-\|x+e\|^2)\exp(-\frac{1}{2}\|x\|^2)\,dx$$

$$= (\frac{1}{2\pi})^{\frac{d}{2}}\exp(-\|e\|^2)\int_{\mathbb{R}^d} x\exp(-\frac{3}{2}\|x\|^2 - 2x^Te)\,dx$$

$$= (\frac{1}{2\pi})^{\frac{d}{2}}\exp(-\frac{1}{3}\|e\|^2)\int_{\mathbb{R}^d} x\exp(-\frac{3}{2}\|x+\frac{2}{3}e\|^2)\,dx$$

$$= (\frac{1}{3})^{\frac{d}{2}}\exp(-\frac{1}{3}\|e\|^2)\int_{\mathbb{R}^d} x\cdot\mathcal{N}(-\frac{2}{3}e,\frac{1}{3}I)\,dx$$

$$= (\frac{1}{3})^{\frac{d}{2}}\exp(-\frac{1}{3}\|e\|^2)\int_{\mathbb{R}^d} x\cdot\mathcal{N}(-\frac{2}{3}e,\frac{1}{3}I)\,dx$$

$$= (\frac{1}{3})^{\frac{d}{2}}\exp(-\frac{1}{3}\|e\|^2)\cdot-\frac{2}{3}e$$

$$= -2(\frac{1}{3})^{\frac{d}{2}+1}\exp(-\frac{1}{3}\|e\|^2)e$$

Additionally we consider $\mathbb{E}[\exp(-\|x+e\|^2)e]$

$$\mathbb{E}[\exp(-\|x+e\|^2)e] = (\frac{1}{2\pi})^{\frac{d}{2}}\int_{\mathbb{R}^d} e\exp(-\|x+e\|^2)\exp(-\frac{1}{2}\|x\|^2)\,dx$$

$$= (\frac{1}{2\pi})^{\frac{d}{2}}\exp(-\|e\|^2)\int_{\mathbb{R}^d} e\exp(-\frac{3}{2}\|x\|^2 - 2x^Te)\,dx$$

$$= (\frac{1}{2\pi})^{\frac{d}{2}}\exp(-\frac{1}{3}\|e\|^2)\int_{\mathbb{R}^d} e\exp(-\frac{3}{2}\|x+\frac{2}{3}e\|^2)\,dx$$

$$= (\frac{1}{3})^{\frac{d}{2}}\exp(-\frac{1}{3}\|e\|^2)\int_{\mathbb{R}^d} e\cdot\mathcal{N}(-\frac{2}{3}e,\frac{1}{3}I)\,dx$$

$$= (\frac{1}{3})^{\frac{d}{2}}\exp(-\frac{1}{3}\|e\|^2)\int_{\mathbb{R}^d} e\cdot\mathcal{N}(-\frac{2}{3}e,\frac{1}{3}I)\,dx$$

$$= (\frac{1}{3})^{\frac{d}{2}}\exp(-\frac{1}{3}\|e\|^2)\cdot e$$

$$= (\frac{1}{3})^{\frac{d}{2}}\exp(-\frac{1}{3}\|e\|^2)e$$

Putting everything together, we get the following:

$$- 2\theta_2\mathbb{E}[\exp(-\|x+e\|^2)(\beta^Tx - \theta_1^T(x+e))]$$

$$= -2\theta_2\beta^T\mathbb{E}[\exp(-\|x+e\|^2)x] + 2\theta_2\theta_1^T\mathbb{E}[\exp(-\|x+e\|^2)x] + 2\theta_2\theta_1^T\mathbb{E}[\exp(-\|x+e\|^2)e]$$

$$= -2\theta_2\beta^T(-2(\frac{1}{3})^{\frac{d}{2}+1}\exp(-\frac{1}{3}\|e\|^2)e) + 2\theta_2\theta_1^T(-2(\frac{1}{3})^{\frac{d}{2}+1}\exp(-\frac{1}{3}\|e\|^2)e) + 2\theta_2\theta_1^T((\frac{1}{3})^{\frac{d}{2}}\exp(-\frac{1}{3}\|e\|^2)e)$$

$$= 4\theta_2\beta^T((\frac{1}{3})^{\frac{d}{2}+1}\exp(-\frac{1}{3}\|e\|^2)e) - 4\theta_2\theta_1^T((\frac{1}{3})^{\frac{d}{2}+1}\exp(-\frac{1}{3}\|e\|^2)e) + 2\theta_2\theta_1^T((\frac{1}{3})^{\frac{d}{2}}\exp(-\frac{1}{3}\|e\|^2)e)$$

$$= 2\theta_2((\frac{1}{3})^{\frac{d}{2}+1}\exp(-\frac{1}{3}\|e\|^2)(\theta_1^Te + 2\beta^Te))$$

We then go on to evaluate $\theta_2^2\mathbb{E}[\exp(-2\|x+e\|^2)]$ using the same calculation method as above, and find that $\theta_2^2\mathbb{E}[\exp(-2\|x+e\|^2)] = \theta_2^2((\frac{1}{5})^{\frac{d}{2}}\exp(-\frac{2}{5}\|e\|^2))$
Putting everything together we find that the loss of the model is:

$$f_l(e,\theta) = \beta^T\beta - 2\beta^T\theta_1 + \theta_1^T\theta_1 + \theta_1^Tee^T\theta_1 + 2\theta_2((\frac{1}{3})^{\frac{d}{2}+1}\exp(-\frac{1}{3}\|e\|^2)(\theta_1^Te + 2\beta^Te)) + \theta_2^2((\frac{1}{5})^{\frac{d}{2}}\exp(-\frac{2}{5}\|e\|^2)).$$

We take the derivative with respect to $\theta_1$ and we find that the loss' derivative is:

$$-2\beta + 2\theta_1 + 2ee^T\theta_1 + 2\theta_2((\frac{1}{3})^{\frac{d}{2}+1}\exp(-\frac{1}{3}\|e\|^2)e$$

Solving for $\theta_1$ after equating the derivative to zero, we find that $\theta_1$ is

$$\theta_1 = (I - \frac{ee^T}{1 + \|e\|^2})(\beta - \theta_2((\frac{1}{3})^{\frac{d}{2}+1} \exp(-\frac{1}{3}\|e\|^2)e)$$

Similarly, we take the derivative with respect to $\theta_2$ and we find it to be:

$$2((\frac{1}{3})^{\frac{d}{2}+1} \exp(-\frac{1}{3}\|e\|^2)(\theta_1^T e + 2\beta^T e)) + 2\theta_2((\frac{1}{5})^{\frac{d}{2}} \exp(-\frac{2}{5}\|e\|^2))$$

setting the derivative to zero and solving for $\theta_2$ we find that $\theta_2$ is:

$$\theta_2 = \frac{-(\frac{1}{3})^{\frac{d}{2}+1} \exp(-\frac{1}{3}\|e\|^2)(\theta_1^T e + 2\beta^T e)}{(\frac{1}{5})^{\frac{d}{2}} \exp(-\frac{2}{5}\|e\|^2)}$$

From this point on we make the assumption that $e$ is in the direction of $\beta$ (i.e., $e = k\frac{\beta}{\|\beta\|}$) for some $k \in \mathbb{R}$. As such, note that $\|e\| = k$. For simplification and ease of computation, we make the following notational substitutions:

$$(\frac{1}{3})^{\frac{d}{2}+1} \exp(-\frac{1}{3}k^2) = m$$

$$(\frac{1}{5})^{\frac{d}{2}} \exp(-\frac{2}{5}k^2) = y$$

Re-writing the expressions of $\theta_1$ and $\theta_2$ we get:

$$\theta_1 = (I - \frac{ee^T}{1 + \|e\|^2})(\beta - \theta_2((\frac{1}{3})^{\frac{d}{2}+1} \exp(-\frac{1}{3}\|e\|^2)e)$$

$$= (I - \frac{ee^T}{1 + \|e\|^2})(\beta - \theta_2 me)$$

$$\theta_2 = \frac{-m(\theta_1^T e + 2\beta^T e)}{y}$$

We go on to simplify the expressions for $\theta_1$ and $\theta_2$:

$$\theta_1 = (I - \frac{ee^T}{1 + \|e\|^2})(\beta - \theta_2 me)$$

$$= (I - \frac{ee^T}{1 + \|e\|^2})(\beta + \frac{m(\theta_1^T e + 2\beta^T e)}{y}me)$$

$$= (I - \frac{ee^T}{1 + \|e\|^2})(\beta + \frac{m^2(\theta_1^T e + 2\beta^T e)}{y}e)$$

$$\theta_1 - (I - \frac{ee^T}{1 + \|e\|^2})\frac{m^2(\theta_1^T e + 2\beta^T e)}{y}e = (I - \frac{ee^T}{1 + \|e\|^2})\beta$$

$$\theta_1 - (I - \frac{ee^T}{1 + \|e\|^2})\frac{m^2\theta_1^T e}{y}e = (I - \frac{ee^T}{1 + \|e\|^2})\beta + (I - \frac{ee^T}{1 + \|e\|^2})\frac{2m^2\beta^T e}{y}e$$

$$\theta_1 - (\frac{1}{1 + \|e\|^2})\frac{em^2\theta_1^T e}{y}e = (I - \frac{ee^T}{1 + \|e\|^2})\beta + (\frac{1}{1 + \|e\|^2})\frac{2em^2\beta^T e}{y}$$

We let $z = -(\frac{1}{1+k^2})\frac{m^2}{y}$ and realize that:

$$(I + zee^T)\theta_1 = (I - \frac{ee^T}{1 + \|e\|^2})\beta + (\frac{1}{1 + \|e\|^2})\frac{2em^2\beta^T e}{y}$$

$$(I + zee^T)\theta_1 = (I - \frac{ee^T}{1 + \|e\|^2})\beta + 2\frac{m^2}{y}\frac{ee^T}{1 + \|e\|^2}\beta$$

$$(I + zee^T)\theta_1 = \beta - \frac{ee^T}{1 + \|e\|^2}\beta + 2\frac{m^2}{y}\frac{ee^T}{1 + \|e\|^2}\beta$$

We make use of the substitution $e = k\frac{\beta}{\|\beta\|}$

$$(I + zee^T)\theta_1 = \beta - \frac{k^2}{1+k^2}\beta + 2\frac{m^2}{y}\frac{k^2}{1+k^2}\beta$$

$$(I + zee^T)\theta_1 = \beta(\frac{1}{1+k^2})(1 + 2\frac{m^2}{y}k^2)$$

We invert the left side using the Sherman Morrison formula:

$$\theta_1 = (I - \frac{zee^T}{1 + z\|e\|^2})\beta(\frac{1}{1+k^2})(1 + 2\frac{m^2}{y}k^2)$$

$$\theta_1 = (I - \frac{zee^T}{1 + z\|e\|^2})\beta(\frac{1}{1+k^2})(1 + 2\frac{m^2}{y}k^2)$$

$$\theta_1 = \beta(\frac{1}{1+zk^2})(\frac{1}{1+k^2})(1 + 2\frac{m^2}{y}k^2)$$

We simplify this by letting $(\frac{1}{1+zk^2})(\frac{1}{1+k^2})(1 + 2\frac{m^2}{y}k^2) = c$ and thus $\theta_1 = \beta c$. We now substitute this expression back to find the expression of $\theta_2$:

$$\theta_2 = \frac{-m(\theta_1^T e + 2\beta^T e)}{y}$$

$$\theta_2 = \frac{-m(c\beta^T e + 2\beta^T e)}{y}$$

$$\theta_2 = \frac{-m}{y}(c\beta^T e + 2\beta^T e)$$

$$\theta_2 = \frac{-m}{y}(\beta^T e)(2 + c)$$

$$\theta_2 = \frac{-m}{y}k\|\beta\|(2 + c)$$

We simplify the expression for $\theta_2$ as well by noting that $\theta_2 = p\|\beta\|$ where $p = \frac{-m}{y}k(2 + c)$ We now calculate the loss of the strategic agent:

$$f_e(\theta, e) = \mathbb{E}[\theta_1^T(x + e) + \theta_2 \exp(-\|x + e\|^2)]$$

$$= \theta_1^T e + \theta_2(\frac{1}{3})^{\frac{d}{2}}\exp(-\frac{1}{3}\|e\|^2)$$

$$= \theta_1^T e + \theta_2 3(\frac{1}{3})^{\frac{d}{2}+1}\exp(-\frac{1}{3}\|e\|^2)$$

$$= \theta_1^T e + \theta_2 3m$$

$$= c\beta^T e + 3mp\|\beta\|$$

$$= ck\|\beta\| + 3mp\|\beta\|$$

$$= \|\beta\|(ck + 3mp)$$

We then now re-evaluate the loss of the model player in terms of the simplified expressions we have found.

$$f_l(e, \theta) = \beta^T\beta - 2\beta^T\theta_1 + \theta_1^T\theta_1 + \theta_1^T ee^T\theta_1 +$$

$$2\theta_2((\frac{1}{3})^{\frac{d}{2}+1}\exp(-\frac{1}{3}\|e\|^2)(\theta_1^T e + 2\beta^T e)) + \theta_2^2((\frac{1}{5})^{\frac{d}{2}}\exp(-\frac{2}{5}\|e\|^2))$$

$$= \beta^T\beta - 2\beta^T\theta_1 + \theta_1^T\theta_1 + \theta_1^T ee^T\theta_1 + 2\theta_2(m(\theta_1^T e + 2k\|\beta\|)) + \theta_2^2 y$$

$$= \|\beta\|^2 - 2c\beta^T\beta + c^2\beta^T\beta + c^2\beta^T ee^T\beta + 2p\|\beta\|(m(c\beta^T e + 2k\|\beta\|)) + p^2\|\beta\|^2 y$$

$$= \|\beta\|^2 - 2c\|\beta\|^2 + c^2\|\beta\|^2 + c^2k^2\|\beta\|^2 + 2pmck\|\beta\|^2 + 4pmk\|\beta\|^2 + p^2 y\|\beta\|^2$$

$$= (1 - 2c + c^2 + c^2k^2 + 2pmck + 4pmk + p^2 y)\|\beta\|^2$$

We assume that $d = 2$ and we consider the loss for the model player with varying values of $k$. Optimizing over this, we see that in the small model setting, the agent is incentivized to use the value of $k = 1$, which corresponds to $\frac{\beta}{\|\beta\|}$. This then gives the model a loss of $\frac{1}{2}\|\beta\|^2$. However, in the larger model case, the agent is incentivized to give a value of $k$ of $\approx 3.4$. This results in a higher model loss of $\approx 0.78\|\beta\|^2$. Figure 2 shows the learner plots.

# E  Further details on the Multi-Agent RL Example

Our procedure for constructing the Markov follows from a couple of foundational principles. Given that we are in a two-player game with players $A$ and $B$, we make payoff matrices that make one action for player $A$ the dominant strategy across all states (e.g., our example in 3. We choose the action 0). It is important to note that though a particular strategy is dominant across states, it does not mean that player $A$ will have the same payoff across all these states. We then make all the transitions entirely independent of this player $A$'s actions. With this, we then design the payoff matrices for player $B$ to be such that depending on how much weight the player $A$ puts on action 0 ($p$), they are incentivized to move to another state.

To do this concretely, we first instantiate a number of states and corresponding thresholds for which the player $B$ would be incentivized to transition from one state to the next. We then use Nash $Q$ learning [50] to find what values of player $B$'s payoff matrix would result in behavior that is such that the Nash policy for player $B$ below some threshold has them preferring, for example, moving to the next state but above this threshold they would prefer staying in the current state.

