# OpenReview forum: "Understanding Model Selection for Learning in Strategic Environments"
_NeurIPS.cc/2024/Conference — NeurIPS 2024 poster_

### Official Review · Reviewer_8iV5 · 2024-07-10

**Soundness:** 4
**Presentation:** 3
**Contribution:** 2
**Rating:** 5
**Confidence:** 3

**Summary:**

The paper studies (non-)monotonicity of equilibrium payoff in certain classes of two-player games, which has implications for strategic machine learning.  Under structural assumptions, the main results are: (1) if the unique equilibrium is not Pareto-optimal, then a player can unilaterally restrict the action space and obtain a better equilibrium in the new, restricted game (similar phenomena happen in more specific example games corresponding to strategic machine learning), and (2) there's an algorithm for selecting the best "model class" (i.e., action subspace) in the same setting.

**Strengths:**

The authors identify an interesting (and to some extent, realistic) phenomenon and establish formal claims about it.  The paper is well written and polished.  The message might be interesting to practitioners.

**Weaknesses:**

My main complaints are (1) the model is a bit unusual and idealized (e.g., assuming uniqueness of equilibrium) for strategic machine learning, and (2) the results don't appear proportionally strong.  See detailed comments below.

**Questions:**

(also including detailed comments)

Line 50: extra space after "--"

Quick comments on contributions (before reading sec 1.1 and anything after that): it sounds like you are treating strategic machine learning as a simultanous-move game, rather than a Stackelberg game, which is a bit unusual.  Is there a reason for that (for one thing, I suppose your results won't hold in the Stackelberg game formulation, where "larger models" are always no worse...)?  Also it sounds like the results are really about abstract games rather than specific games induced by machine learning tasks, which makes me wonder to what extent the paper is about model selection (and not strategic interaction in general).

Around line 96: well, I'm not sure "papers ... all analyze games in which a learner attempts to update their model through repeated retraining".  In particular, there's a bunch of papers in strategic machine learning that study one-shot solutions of the Stackelberg game, including the seminal paper [10] which the authors cite repeatedly.  I also wouldn't view these papers as about learning in games.

After reading sec 3: I still feel the results are somewhat ambivalent.  In particular, if the learner has the commitment power and information to "select a good model" in the simultaneous-move setting, then why can't the learner simply play a Stackelberg equilibrium?  Of course this is a very vague argument, but so is the one presented in the paper...

Line 304: extra space

Sec 4: correct me if I'm wrong -- is the result simply saying "try every class and pick the best one"?  And I imagine you can remove the doubling procedure if there's a target suboptimality that can be tolerated?

**Limitations:**

No concerns.

---

> ### Author Rebuttal · Authors · 2024-08-05
>
> Firstly, we would like to thank the reviewer for the time and effort spent looking through our work. We are delighted to see how the reviewer recognizes this research direction as interesting and points to it as a realistic phenomenon that has potential interest to practitioners. Additionally, we appreciate the comment on the paper's writing. Please find below our expansion on some of the comments and questions raised:
>
> **Punctuations and editorial concerns**
>
> Thank you for the comments on punctuation edits. We will correct these in the final version.
>
> **...I also wouldn't view these papers as about learning in games.**
>
> We concede that not all papers referenced take the lens of learning in games; however, they are all about deploying learning algorithms/classifiers in game-theoretic environments. In our work, we show that model complexity in these environments is a non-trivial matter over and beyond simply generalization error considerations.
>
>
> **The model is a bit unusual and idealized (e.g., assuming uniqueness of equilibrium) for strategic machine learning**
>
> As a comment, our *general model does* **not** make assumptions on uniqueness of equilibrium.  We do make assumptions for the proof of a negative result (please see general comment for more information on assumptions).
>
> **It sounds like you are treating strategic machine learning as a simultanous-move game, rather than a Stackelberg game**
>
> Our model also does indeed capture cases where the interaction between the learner and environment is a Stackelberg game as well. We describe our findings in the preliminaries section and point to the Appendix to the full set of results including illustrations for Stackelberg environments. Concretely: in the paper, we look at how performance at equilibrium scales as a function of model class expressivity across 4 types of strategic environments and find that:
>
> 1. In Stationary, and Stackelberg environments where the learner leads, performance at equilibrium scales monotonically as a function of model class expressivity. This is in line with the view of traditional machine learning where performance does scale monotonically as a function of the model class’ expressivity.
> 2. In Stackelberg environments where the learner follows as well as in Nash environments, performance at equilibrium does **NOT** scale monotonically as a function of model class expressivity. This is in stark contrast to the growing consensus that, the larger and more expressive the model, the better performance
>
> **I still feel the results are somewhat ambivalent. In particular, if the learner has the commitment power and information to "select a good model" in the simultaneous-move setting, then why can't the learner simply play a Stackelberg equilibrium?**
>
> The problem that the learner faces is that of first selecting a class of models (which can be thought of as an action space) and the from that, when engaging in a game with the environment selecting a particular model (or action) for the game. The learner thus has “commitment power” in that they are first able to select a class of models or space over which to play. This work shows that this first task of selecting a class of models over which to play, depending on the type of game, is non-trivial. In particular, for Nash games and Stackelberg games where they are the follower, it very well may be the case that increasing the size of your model class may lead to equilibria that yield worse payoffs for the learner. This is in direct contradiction to conventional wisdom in machine learning which seems to suggest that adding models / actions to a set which you are optimizing over will lead to performance that is at least as good as when optimizing over a strict subset.
>
> We consider performance at equilibrium positions in each of the sub-game archetypes as these are stationary points. Considering performance at equilibrium is a common benchmark in strategic interactions’ analyses e.g., [1][2]. in a Nash game, therefore, a Stackelberg equilibrium may not be a Nash equilibrium and therefore would not be a stable point for that particular game archetype.
>
> **Sec 4: correct me if I'm wrong -- is the result simply saying "try every class and pick the best one"? And I imagine you can remove the doubling procedure if there's a target suboptimality that can be tolerated?**
>
> Please refer to the general comment.
>
> **References**
>
> [1] Meena Jagadeesan, Michael Jordan, Jacob Steinhardt, and Nika Haghtalab. Improved bayes risk can yield reduced social welfare under competition
>
> [2]Juan Perdomo, Tijana Zrnic, Celestine Mendler-Dünner, and Moritz Hardt. “Performative prediction.”

---

> > ### Comment · Reviewer_8iV5 · 2024-08-08
> >
> > Thank you for your response, which is quite helpful.  To be honest, I still have some concerns, but I also realize these concerns are mostly subjective.  Given that I will increase my score to 5, leaning towards acceptance.

---

> > > ### Author Response · Authors · 2024-08-09
> > >
> > > Thank you for your consideration and adjustment.

---

### Official Review · Reviewer_RwJQ · 2024-07-12

**Soundness:** 3
**Presentation:** 3
**Contribution:** 3
**Rating:** 6
**Confidence:** 3

**Summary:**

They study the trade-off between model expressivity and performance at equilibrium in presence of strategic interactions. They show that strategic interactions can cause non-monotone performance at equilibrium when the model gets more expressive.

They show Braess'-paradox like examples where reverse scaling occurs, i.e. the larger and more expressive the model class a learner optimizes over, the lower their performance at equilibrium.
Furthermore, they formulate a problem of model-selection in games. In this formulation, each player has a number of action sets to choose from and must find one that yields the best payoff. Under the assumption that both environment and the learner use SGD, they propose an algorithm for this problem where the avg payoff over iterations in their algorithm concentrates to the payoff in the Nash equilibrium.

**Strengths:**

I think overall the results are interesting.

**Weaknesses:**

Overall, I found the results interesting.

**Questions:**

Assumption 4.1, what is F?

Thm 3.4. define \delta^* and e^*.

Line 149: define \Omega and \calE.

**Limitations:**

same as above.

---

> ### Author Rebuttal · Authors · 2024-08-05
>
> We would like to thank the reviewer for the time and effort spent reviewing our work. We are delighted to see that the reviewer recognizes the importance of the findings discussed. Please find below our response to your questions and concerns.
>
> **Questions**
>
> Thank you for pointing out places where we could further describe the mathematical notation we used.
>
> **Assumption 4.1, what is F?**
>
> Here F is the gradient operator (i.e., F(x) is the gradient evaluated at point x).
>
> **Thm 3.4.**
>
> Here $\theta^*, e^*$ is the unique Nash equilibrium in $\Theta \times \mathcal{E}$.
>
> **Line 149: define $\Omega$ and $\mathcal{E}$.**
>
> Here $\Omega$ is the space of all model classes and $\mathcal{E}$ is the space of all actions the environment can select.

---

> > ### Comment · Reviewer_RwJQ · 2024-08-12
> >
> > Thanks for your response.
> >
> > I have one more comment regarding the technical details, I find Example 2 for showing these Braess-paradox type of results simple and not surprising. Please correct me if I am wrong here. But it just says that if there are some features that the agents don't want to be used in the prediction model, they will add noise to those features, and then the prediction model is better off not using those features. Am I missing something here?

---

> > > ### Author Response · Authors · 2024-08-13
> > >
> > > Thank you for your question:
> > >
> > > As a general comment on the examples we selected, these examples were selected to highlight the breadth of scenarios where the Braess-paradox-like phenomenon exhibits itself. The hope was to put together cases from different facets of ML to show the pervasiveness of this phenomenon.
> > >
> > > Example 2, in particular, was interesting to us both in its connection to previous work done (e.g., performative prediction) as well as in the implications it has for practitioners. For us, the connection that privacy-preserving / fairness-inducing techniques could be argued for from a utility maximization lens for the learner was an interesting takeaway that we thought ought to be highlighted and added nuance to the development of robust real-world machine learning tools. We will make sure to highlight this connection and emphasize the salience of this example better in a future version.
> > >
> > > As a side note, we also see in Example 3 (in the appendix due to space considerations) that this phenomenon not only shows up when the learner uses new, different sets of features but also when they consider more complex models that make use of the same features but with more complex functions to process the information (e.g., Neural Networks with more parameters).

---

### Official Review · Reviewer_QRet · 2024-07-14

**Soundness:** 3
**Presentation:** 2
**Contribution:** 3
**Rating:** 6
**Confidence:** 3

**Summary:**

This paper studies the relationship between model class expressivity and equilibrium performance when there are strategic interactions between agents in MARL settings. In contrast with the conventional scaling laws in machine learning, where task performance typically improves with larger or more expressive model classes, this paper highlights a phenomenon similar to Braess' paradox where in certain strategic environments like 2-player games between the learner and the environment with a unique Nash equilibrium, using less expressive model classes for the learner can lead to better equilibrium outcomes. It is theoretically proved that if the Nash equilibrium is not Pareto-optimal, the learner can always restrict its model class to achieve better outcomes. This is further explained with illustrative examples for a 2-player Markov game and for strategic classification in Participation Dynamics. Finally, the authors formulate the problem of model selection in strategic environments and propose a successive elimination based algorithm for the learner to identify the best model class whose Nash equilibrium yields the highest payoff among a candidate set of model classes.

Overall, the main ideas presented in this paper are:
- The choice of model class should be treated as a strategic action when deploying models in strategic environments.
- There's a need to rethink scaling laws before deploying increasingly complex models in real-world strategic settings.

**Strengths:**

1. This paper challenges the conventional wisdom about scaling laws in machine learning, and draws attention to how strategic interactions between the learner and its environment can affect equilibrium performance. This is an important and relevant topic towards making machine learning models robust for real world applications.

2. The paper clearly outlines the assumptions made in the different settings for which theoretical guarantees  have been provided, along with illustrative examples in related domains to better understand the applicability of its insights.

**Weaknesses:**

1. Some of the theoretical results presented in the paper rely on strong assumptions, eg. strong monotonicity ensuring existence of a unique Nash equilibrium, or assuming the availability of SGD estimators with decreasing step sizes for all players, which may not always hold in practice. This paper does not focus on empirical evaluations to validate its claims.

2. The proposed algorithm for online model selection shows the existence of a tractable algorithm to choose between candidate model classes under certain assumptions, but the number of interactions required with the environment increases with the size of the candidate set $n$ (Proposition 4.3) which would be computationally expensive for large model classes.

3. The illustrative examples focus on simplified policy classes which may not be representative of practical applications, and the effect of different model architectures or generalization to larger team sizes is not considered.

**Questions:**

1. Line 221: "... for some $\bar{p}\in [0,1]$" - should this be $\bar{p}>=0.5$ since "$p\in [1-\bar{p}, \bar{p}]$" does not seem to make sense otherwise?

2. Would the analysis presented in this paper also extend to repeated games where player strategies can be adaptive and the effect of model selection might be time dependent?

**Limitations:**

Yes, the authors describe the limitations of this approach and potential directions for future work.

---

> ### Author Rebuttal · Authors · 2024-08-05
>
> We firstly would like to thank the reviewer for the time and effort spent in looking through our work. We are delighted to see how the reviewer recognizes how this work challenges conventional wisdom with respect to model selection in machine learning. Additionally, we appreciate the comment on the relevance of this line of work in building robust machine learning models for real world applications. Please find below our expansion on some of the comments and questions you raised:
>
> **Weaknesses**
>
> **1. Some of the theoretical results presented in the paper rely on strong assumptions, e.g., strong monotonicity ensuring the existence of a unique Nash equilibrium or assuming the availability of SGD estimators with decreasing step sizes for all players, which may not always hold in practice. This paper does not focus on empirical evaluations to validate its claims.**
>
> **2.The illustrative examples focus on simplified policy classes, which may not be representative of practical applications. The effect of different model architectures or generalization to larger team sizes is not considered.**
>
> **3.The proposed algorithm for online model selection shows the existence of a tractable algorithm to choose between candidate model classes under certain assumptions, but the number of interactions required with the environment increases with the size of the candidate set $n$**
>
> For the main theorem (Theorem 3.4.) and the work on the identification of a model class please see the general comment.
>
> We are also excited by the prospect of future work showing this phenomenon clearly through real-world large scale controlled experiments and deployments which we did not have the capacity to do. Our examples were selected to illustrate the range of scenarios one can expect to observe non-monotonicity of performance degradation–not an empirical evaluation.
>
> **Questions**
>
> **Line 221: "... for some p¯∈[0,1]" - should this be p¯>=0.5 since "p∈[1−p¯,p¯]" does not seem to make sense otherwise?**
>
> Yes, you are correct. Thank you for pointing this out. We will edit this in the final version.
>
> **Would the analysis presented in this paper also extend to repeated games where player strategies can be adaptive and the effect of model selection might be time dependent?**
>
> We believe that we would be able to see the same phenomenon present itself in more complicated regimes, such as repeated game settings with complicated strategies or time-adaptive policy classes. We anticipate some of the analyses would need to be adjusted to take into account additional complexity, and we look forward to future work that dives more deeply into the nuances of more complicated games and strategies.

---

> > ### Comment · Reviewer_QRet · 2024-08-13
> > **Acknowledgement**
> >
> > Thank you for the response. I will maintain my original score, leaning towards acceptance.

---

### Official Review · Reviewer_yWQz · 2024-07-14

**Soundness:** 4
**Presentation:** 4
**Contribution:** 3
**Rating:** 7
**Confidence:** 2

**Summary:**

The authors study a strategic learning setting formalized as a game involving a player whose action space is some function class that the player optimizes over. The paper focuses on theoretically demonstrating that, in such games, a learning agent may have an incentive to unilaterally commit to a restricted action space.

The main theoretical result of the paper appears to be Theorem 3.4 which states that in unconstrained strongly monotone games with inefficient equilibria, a player can always strictly improve its utility at equilibrium by gaining commitment power in some fashion. The authors also describe the sample complexity of using successive elimination to identify a model class for which the learner's equilibrium utility is near optimal.

**Strengths:**

The paper's study of the impact of model selection on equilibrium performance is to the best of my knowledge novel; usually commitment is studied in the context of commiting to a single action rather than a set-valued commitment more remiscient of meta-games. Though it is intuitive that set-valued commitment should improve one's equilibrium outcome from the perspective of equilibrium selection, the technical result of the paper (Theorem 3.4) studies when equilibrium is unique and thus commitment has the effect of entirely changing the equilibrium set rather than effecting favorable equilibrium selection, which seems less obvious. It seems that such a result should not be possible when model selection coincides with usual commitment where model selection is only of singleton classes, though it would be helpful for the authors to clarify whether this is the case. Generally, the writing of the paper is polished and accessible.

**Weaknesses:**

* Restricting one's model class seems to be equivalent to a form of set-valued commitment. From that perspective, it does not seem particularly surprising that gaining commitment power improves one's equilibrium outcome. If this intuition is not correct and there's subtlety, the authors should clarify so in the paper, which currently does not really discuss model selection as commitment.

* The appendix proofs could be written more clearly. In the Theorem 3.4 proof, how is the possibility that $\nabla_\theta BR_e(\theta^*) = -1$ being ruled out?

* Could the authors clarify why section 4 is titled "Online learning..."---it's not obvious to me what the online learning aspect of the problem is. It seems to still consider a setting with a fixed game; perhaps the authors meant game dynamics rather than online learning?

**Questions:**

See weaknesses section.

**Limitations:**

The authors have addressed any potential negative impacts.

---

> ### Author Rebuttal · Authors · 2024-08-05
>
> Firstly, we would like to thank the reviewer for the time and effort spent looking through our work. We are delighted to see how the reviewer recognizes the novelty of the paper’s study as well as the unintuitive, surprising, yet impactful consequences of the notions investigated. Additionally, we appreciate the comment on the paper's writing. Please find below our expansion on some of the comments and questions you raised:
>
>
> **Restricting one's model class seems to be equivalent to a form of set-valued commitment. From that perspective, it does not seem particularly surprising that gaining commitment power improves one's equilibrium outcome. If this intuition is not correct and there's subtlety, the authors should clarify so in the paper, which currently does not really discuss model selection as commitment**
>
> You are correct in identifying commitment as a crucial factor in giving rise to this phenomenon. However, the non-monotonicity of performance we illustrate does not arise from gaining or losing commitment power (in our model, model class selection occurs at the beginning regardless of the type of game). Non-monotonicity arises from the difficulty in selecting a commitment in the presence of strategic environments.
>
> As to the question of why it is we observe this complexity in the model class expressivity to equilibrium-payoff function, we make links to work that has explored this unintuitive landscape. Most prominently, Braess’ paradox made the observation of how adding a road to a road network can slow down traffic flow through the network. One can view our results as an instantiation of this Braess paradox-like phenomenon within the realm of machine learning. Other works include results on the non-convexity of Stackelberg losses [1]
>
> **The appendix proofs could be written more clearly. In the Theorem 3.4 proof, how is the possibility that ∇θBRe(θ∗)=−1 being ruled out?**
>
> Thank you for your feedback on the appendix, we will work to improve clarity in this section.
> For Theorem 3.4., This is another regularity assumption on the game for this negative result. For a more concrete discussion, please see general comment.
>
> **Could the authors clarify why section 4 is titled "Online learning..."---it's not obvious to me what the online learning aspect of the problem is. It seems to still consider a setting with a fixed game; perhaps the authors meant game dynamics rather than online learning?**
>
> Please see general comment for discussion
>
> [1] Basar, T., & Olsder, GJ. (1999). Dynamic noncooperative game theory. 2nd ed

---

> > ### Comment · Reviewer_yWQz · 2024-08-09
> > **Response**
> >
> > Thanks for the clarifications---and I agree that non-monotonicity is non-obvious from the perspective of commitment. I maintain my positive review.

---

### Author Rebuttal · Authors · 2024-08-05

We would like to thank all the reviewers for their input to our paper as well as their comments. We appreciate the broad consensus and recognition of the importance of this research avenue within the Machine Learning community, particularly its salience in ensuring the development of robust models for real world applications. We believe that the issue that we highlight is important to characterize and study since it can give us insights into the performance of ML algorithms in real-world environments. Below are some comments to help with any lingering issues:

**On the assumptions made:**

For Theorem 3.4. we would like to note that this result is a negative result which is an existence proof. In particular, we show that even under strong assumptions on the game (e.g., strong monotonicity), we are able to observe non-monotonicity. Through our examples, we show how when these assumptions are relaxed in games without some of the nice properties of the theorem this phenomenon still presents itself. The multi-agent reinforcement learning example we provide, for instance, illustrates this as we have a Nash game where strong monotonicity does not hold, yet we find that performance does not scale monotonically with respect to the expressivity of the policy class. What we sought to show through the theorem was the pervasiveness of this phenomenon even in games with very strong assumptions on their structure.

**On Online learning for Model Selection in Games:**

Section 4 follows from the non-convexity of the loss as well as the realization that one cannot know what model class to commit to without knowing the objective of the environment. Learning this objective can only be done through repeated interaction with the environment—i.e., a form of online learning. This section provides a framework and first algorithm for thinking about the optimization problem that a learner has to solve. We look forward to future work that looks more expansively at this domain. As posed, with no assumption on the structure of the equilibrium payoff’s relationships between model classes, the learner inevitably has to try out all model classes and thus the dependence on $n$. We leave for future work the question of whether it possible to be no-regret within and across model classes. Furthermore work that is able to better characterize and take advantage of the structure of the non-monotonic landscape could aide in going beyond the dependence on $n$.

We would like to reiterate our appreciation of the general positive reception of the work. We look forward to engaging further to improve the work and clear up any concerns.

---

### Decision · Program_Chairs · 2024-09-25

**Decision:**

Accept (poster)

**Comment:**

The results in this paper are not as strong as one might like. However, it is a solid piece of work.